# A Bounded Ability Estimation for Computerized Adaptive Testing

**Yan Zhuang**[1,2], **Qi Liu**[1,2,*] **GuanHao Zhao**[1,2], **Zhenya Huang**[1,2],
**Weizhe Huang**[1,2], **Zachary A. Pardos**[3], **Enhong Chen**[1,2], **Jinze Wu**[2,4], **Xin Li**[2,4]

1: Anhui Province Key Laboratory of Big Data Analysis and Application,
University of Science and Technology of China
2: State Key Laboratory of Cognitive Intelligence
3: University of California, Berkeley
4: iFLYTEK Co., Ltd
{zykb,ghzhao0223,hwz871982879,hxwjz}@mail.ustc.edu.cn,
{qiliuql,huangzhy,cheneh,leexin}@ustc.edu.cn, pardos@berkeley.edu

## Abstract

Computerized adaptive testing (CAT), as a tool that can efficiently measure student's ability, has been widely used in various standardized tests (e.g., GMAT and GRE). The adaptivity of CAT refers to the selection of the most informative questions for each student, reducing test length. Existing CAT methods do not explicitly target ability estimation accuracy since there is no student's true ability as ground truth; therefore, these methods cannot be guaranteed to make the estimate converge to the true with such limited responses. In this paper, we analyze the statistical properties of estimation and find a theoretical approximation of the true ability: the ability estimated by full responses to question bank. Based on this, a Bounded Ability Estimation framework for CAT (BECAT) is proposed in a data-summary manner, which selects a question subset that closely matches the gradient of the full responses. Thus, we develop an expected gradient difference approximation to design a simple greedy selection algorithm, and show the rigorous theoretical and error upper-bound guarantees of its ability estimate. Experiments on both real-world and synthetic datasets, show that it can reach the same estimation accuracy using 15% less questions on average, significantly reducing test length.

## 1 Introduction

As the landscape of education is changing rapidly, especially after COVID-19, many schools and institutions move from in-class to online platforms, providing individualized education, such as educational measurement and recommendation. They are looking to "right-size" the learning experience of students according to their ability level [1, 2]. To this end, Computerized Adaptive Testing (CAT) [3] becomes an indispensable tool to efficiently measure student's ability in the areas of standardized testing, computer tutoring, and online courses, through automatically selecting best-suited questions for individual students. Compared with the time-consuming and burdensome paper-and-pencil tests, CAT has been proven to require fewer questions to reach the same measurement accuracy [4, 2].

A typical CAT system is shown in Figure 1: At test step $t$, the Cognitive Diagnosis Model, e.g., Item Response Theory (IRT), as the user model based on psychology, first uses student's previous $t$ item answer responses to estimate his/her current ability $\theta^t$. IRT family has been used for ability estimation in several state assessments, such as OECD/PISA Project [5, 6]. Next, the selection algorithm selects the next item from the entire question bank according to some criteria [7, 8, 9]. Most of them are

---

*Corresponding Author.

37th Conference on Neural Information Processing Systems (NeurIPS 2023).

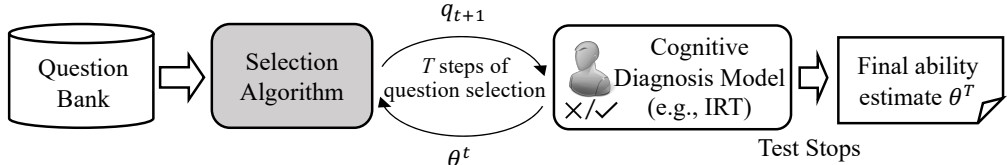

Figure 1: An illustration of the CAT system: At test step $t \in [1, ..., T]$, the selection algorithm uses the current ability estimate $\theta^t$ to select the next question $q_{t+1}$ from the question bank. When the test stops, the $\theta^T$ (i.e., the final estimate of his/her true ability $\theta_0$) will be output.

informativeness metrics such as selecting the question with difficulty closest to his/her current ability estimate $\theta^t$, i.e., the student's probability of answering it correctly is closest to 50% [7]. Obviously, the selection algorithm is the core component to realize CAT's adaptivity and seeks to answer the following question about *accuracy and efficiency*: Can we estimate student's true ability by asking him/her as few questions as possible, with negligible estimation error?

From the perspective of machine learning, CAT can be viewed as a parameter estimation problem with the least cost: it is essentially to select the fewest data samples (questions to be answered) sequentially from the whole unlabeled data (question bank), so that after obtaining their labels (correct/wrong responses), model's hidden parameters (student true ability $\theta_0$) can be accurately estimated. Unfortunately, the exact true ability of student is unknown even to the students themselves, thus it is impossible to find such ground truth in datasets to design/train selection algorithms. As a result, most selection algorithms *are not designed explicitly with the goal of accurate and efficient estimation*. Existing approaches either select representative/diverse items solely from question feature space [9] (deviating from the goal of ability estimation), or require additional training overhead (e.g., Reinforcement Learning-based methods [10, 11, 12, 13]). Although these implicit methods achieve good results in experiments, the theoretical guarantee on approximating student's true ability is also critical for reliable CAT systems especially in standardized tests.

Obviously, the biggest challenge of designing reliable explicit methods is: student's true ability $\theta_0$ is unknown. Therefore, in this work, we propose a general (upper-)Bounded Ability Estimation CAT framework (BECAT), which explicitly targets the accuracy and efficiency of ability estimation. Due to the unknown $\theta_0$, we first find its theoretical approximation $\theta^*$ as the alternative: the ability estimated by his/her full responses on the entire question bank. Hence, our key idea is to select questions such that the estimate can best approximate the ability estimated by full responses. Specifically, we propose an expected gradient difference approximation method based on recent data efficiency/summary technique [14, 15, 16], and design a practical greedy selection algorithm in submodular function, which essentially finds representative items to approximate the gradient of full responses. We further provide the theoretical analysis about its upper-bound of ability estimation error.

To validate BECAT's effectiveness, we conduct experiments on three real-world datasets from different educational platforms. Empirical results show that this simple greedy selection achieves state-of-the-art performance compared with other implicit methods. The main contributions are:

- To better estimate the unknown $\theta_0$, we find its theoretical approximation as the new target for designing an explicit selection algorithm. Based on this, we formally redefine and transform CAT into an adaptive subset selection problem in data summary manner for the first time.

- An effective expected gradient-based selection algorithm is proposed to select appropriate items, which exactly minimizes the estimation error term, therefore admitting theoretical guarantees on ability estimation in CAT systems.

- We show the generality of BECAT — it can be applied to any gradient-based method, including IRT and neural network methods. We observe that BECAT outperforms existing CAT methods at reducing test length, requiring 10%-20% less questions to reach the same estimation accuracy.

## 2  Problem Definitions of CAT

For accurate and efficient assessment, CAT needs to sequentially select best-fitting questions for each student from the question bank $Q$; then uses the corresponding responses for ability estimation.

When the test stops, the final estimate is output as the result/score of this test. The goal of CAT is to accurately estimate examinee's true ability $\theta_0$, while minimizing the number of questions asked [17].

## 2.1 Preliminaries

Specifically, at test step $t \in [1, 2, ..., T]$, given the student's previous $t$ responses $S_t = \{(q_1, y_1), ..., (q_t, y_t)\}$, where $\{q_i\}_{i=1}^{t} \subseteq Q$ are selected sequentially by the selection algorithm and $y$ is the binary outcomes of correct or incorrect; student's current ability can be estimated by minimizing the empirical risk (e.g., binary cross-entropy) from the whole ability space $\Theta$:

$$\theta^t = \arg\min_{\theta \in \Theta} \sum_{i \in S_t} l_i(\theta) = \arg\min_{\theta \in \Theta} \sum_{i \in S_t} -\log p_\theta(q_i, y_i), \tag{1}$$

where $p_\theta(q_i, y_i)$ represents the probability of the response $(q_i, y_i)$ towards a student with $\theta$, and the specific form of $p_\theta$ is determined by IRT. Since the size of $S_t$ is small, Standard Gradient Descent [18, 19] is sufficient to minimize Eq.(1), and requires the computations of $\sum_{i \in S_t} \nabla l_i(\theta)$ — sum of the gradients over the previous $t$ response data. It takes repeated steps in the opposite direction of the gradient, thus leading to a minimum of the empirical risk in Eq.(1).

Next, the selection algorithm selects the next question $q_{t+1}$ from bank $Q$ according to various criteria [7, 8, 12, 13]. The above process will be repeated for $T$ times[2], i.e., $|S| = T$ ($T \leq 20$ in most tests [13]), ensuring the final step estimate $\theta^T$ close to the true $\theta_0$, i.e.,

**Definition 1** (Traditional Definition of CAT). At each step $t$, it will select the most suitable/informative question, according to student's current ability $\theta^t$. When the test ends ($t = T$), the final ability estimate $\theta^T = \arg\min_{\theta \in \Theta} \sum_{i \in S} l_i(\theta)$ can approximate the true ability:

$$\min_{|S|=T} \|\theta^T - \theta_0\|. \tag{2}$$

Unfortunately, directly solving the above optimization problem is infeasible. Because the ground truth ability $\theta_0$ cannot be obtained and, even students themselves cannot know the exact value. As a result, traditional informativeness-based methods [7, 8] use asymptotic statistical properties of Maximum Likelihood Estimation to reduce estimation uncertainty, e.g., selecting the one whose difficulty is closest to student's current ability $\theta^t$; but they are all IRT-specific, i.e., they can not be applied into recent neural networks methods. Although recent active learning-based [9] and reinforcement learning-based [12, 13] methods achieve good experimental results, *there is no evidence that they can theoretically guarantee that estimate can efficiently approach $\theta_0$*, which is unacceptable for CAT systems applied in standardized tests. Testing reliability requires not only satisfactory experimental results, but also good theoretical guarantees [3].

## 2.2 New Definition of CAT

Given that there is no such ground truth $\theta_0$ in the dataset, thus, for designing explicit selection algorithms, we find its approximation as the new target.

**Proposition 1.** *The student's one ability estimate $\theta^*$, estimated by his/her full responses to the entire question bank $Q$, is an approximation of his/her true ability $\theta_0$, that is,*

$$\theta^* \approx \theta_0 \tag{3}$$

*Proof.* When we use consistent estimation approaches, such as Maximum Likelihood Estimation (cross-entropy loss) in Eq.(1), we have $\lim_{t \to \infty} p\left(|\theta^t - \theta_0| \geq \epsilon\right) = 0$, where $t$ is the number of responses (steps) for ability estimation. The size of CAT's question bank is finite (i.e., $t \in [0, |Q|]$) and $\theta^* = \lim_{t \to |Q|} \theta^t$, thus $\theta^* \approx \lim_{t \to \infty} \theta^t \approx \theta_0$, i.e., $\theta^*$ can be regarded as an approximation of $\theta_0$. □

Since this proposition exploits estimator's asymptotic property and may make the approximation not perfect. For example, both the bank size $|Q|$ and various perturbations in student's response

---

[2]In this paper, we only consider the most common fixed-length tests ($T = 20$), and the effect of test length can be found in Appendix E

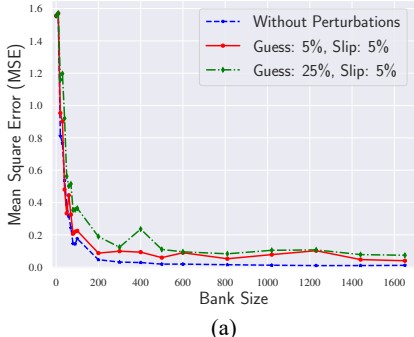
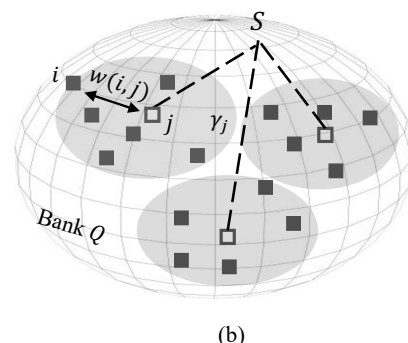

|     |     |
| --- | --- |
| (a) | (b) |

Figure 2: (a) Simulation experiments about Proposition 1 using MSE: $\mathbb{E}[\|\theta^* - \theta_0\|^2]$. In addition to the normal situation (blue), we also show the MSE under different perturbations, for example: Slip5% means that the label has a 5% probability of changing from 1 to 0; Guess25% means that the label changes from 0 to 1 with 25%. (b) The illustration of the optimization problem: selecting subset $S$ to *cover* the whole response data on $Q$. The rectangles represent a student's full responses to the bank $Q$, and $w(i, j)$ measures the similarity of response pair $(i, j)$.

can impact this proposition. Thus, we also conduct simulation experiments to further verify it: we randomly sample 100 $\theta_0$ from $\Theta$ as groundtruth, using the smallest EXAM dataset ($|Q| = 1650$) in Section 4; then use IRT with these $\theta_0$ to simulate the response behavior (correct/wrong) of 100 students. Figure 2(a) shows that when the bank size exceeds 300 ($\approx |Q|/5$), the estimated $\theta^* \approx \theta_0$ (blue). Even if some extreme perturbations (e.g., guess and slip factors [13]) are added, their MSE will not exceed 0.1. Therefore, it is reasonable to replace $\theta_0$ with $\theta^*$ as the ground truth in optimization.

In this way, the selection algorithm can aim to approach $\theta^*$ instead of the unknown $\theta_0$. We can design explicit selection algorithms: *Select a subset of questions $S$ from the bank $Q$, so that the student's ability is estimated only on the subset $S$ while still (approximately) converging to the optimal solution $\theta^*$* (i.e., the estimate that would be obtained if optimizing on the full responses to $Q$). As mentioned in Preliminaries, the ability estimation usually adopts cross-entropy loss with gradient computations, and denote the full gradient of the loss w.r.t. ability parameters by $\sum_{i \in Q} \nabla l_i(\theta)$ — sum of the gradients over full responses. Thus,

**Definition 2** (New Definition of CAT). It will adaptively find a subset $S$ of size $T$ and the corresponding weight $\{\gamma\}_j$ that approximates the full gradient: minimizing their difference for all the possible ability values of the optimization parameter $\theta \in \Theta$ :

$$\min_{|S|=T} \|\theta^T - \theta^*\| \Rightarrow \min_{|S|=T} \max_{\theta \in \Theta} \| \sum_{j \in S} \gamma_j \nabla l_j(\theta) - \sum_{i \in Q} \nabla l_i(\theta)\|. \tag{4}$$

Since we know nothing about the range of student's ability when optimizing, we consider the worst-case approximation error ($\max_{\theta \in \Theta}$) instead of a particular $\theta$. After finding such $S$ and the associated weights $\{\gamma\}_j$, *the gradient updates on $S$ will be similar to the gradient on $Q$ regardless of the value of $\theta$, thus making the estimate close to the target $\theta^*$*. In this way, CAT can be regarded as a subset selection optimization problem in a data-efficiency manner. Also, we find that it is consistent with recent Coreset techniques [20, 15, 21], which approximate the gradients of the full data, so that the model is trained only on the subset while still (approximately) converging to the optimal solution (i.e., the model parameters that would be obtained if training on the full data).

However, compared with the traditional Coreset problem, the biggest technical challenge is: the gradients on bank ($\sum_{i \in Q} \nabla l_i(\theta)$) cannot be calculated without labels. Only the few questions that have been answered in previous steps (i.e., $S_t$) have the corresponding labels. Thus, to simplify the problem, we *assume for the moment that student's full responses are available*. In Section 3.1, we will further propose an expected approximate method to address this.

# 3 The BECAT framework

In this section, to solve the above optimization problem Eq.(4), we design a simple greedy algorithm in submodular functions. More importantly, we provide an upper bound on the expected error of the ability estimate when using our method.

**Optimization.** The above subset selection problem is NP-hard, thus, we transform it based on the recent Coreset method. It proves that the subset $S$ that minimizes the error of estimating the full gradient is upper-bounded by a submodular facility location function that has been used in various summarization applications [22, 23]. Thus

$$\min_{|S|=T} \max_{\theta \in \Theta} \|\sum_{j \in S} \gamma_j \nabla l_j(\theta) - \sum_{i \in Q} \nabla l_i(\theta)\| \Rightarrow \min_{|S|=T} \max_{\theta \in \Theta} \sum_{i \in Q} \min_{j \in S} \|\nabla l_i(\theta) - \nabla l_j(\theta)\|$$

$$\Rightarrow \max_{|S|=T} \sum_{i \in Q} \max_{j \in S} w(i, j), \tag{5}$$

where $w(i, j) \triangleq d - \max_{\theta \in \Theta} \|\nabla l_i(\theta) - \nabla l_j(\theta)\|$ is the gradient similarity between response pair $i = (q_i, y_i)$ and $j = (q_j, y_j)$ for this student. The associated weight of the response $j$, $\gamma_j = \sum_{i \in Q} \mathbf{1}[j = \arg\max_{s \in S} w(i, s)]$, is the number of responses in $Q$ that are most similar to $j \in S$.

Given a subset $S$, $\sum_{i \in Q} \max_{j \in S} w(i, j)$ in Eq.(5) *quantifies the coverage of the whole response data on $Q$*, by summing the similarities $w$ between every $i \in Q$ and its closest item $j \in S$. The semantics of this optimization problem is shown in Figure 2(b). The larger the value of $w(i, j)$, the smaller their gradient difference in ability estimation for all the possible ability $\theta \in \Theta$, which means these two responses $i$ and $j$ have *similar importance/influence on the student's ability estimation*. Thus, the transformed problem in Eq.(5) is equivalent to selecting the most representative responses to form the subset $S$, which shares the same idea (i.e., selecting "representative" items) with previous selection algorithms [12, 9], active learning methods [24, 25] and unsupervised learning [26, 27].

Define a monotone non-decreasing submodular function — the facility location function $F : 2^Q \to \mathbb{R}$: $F(S) = \sum_{i \in Q} \max_{j \in S} w(i, j)$. The submodular optimization provides a near-optimal solution with a $(1 - 1/e)$-approximation bound [28], with simple greedy algorithm for selecting the $t$-th question:

$$q_t = \arg\max_{(q,y) \in Q \setminus S_{t-1}} \Delta((q, y)|S_{t-1}). \tag{6}$$

where $\Delta((q, y)|S_{t-1}) = F(\{(q, y)\} \cup S_{t-1}) - F(S_{t-1})$ and $S_{t-1}$ is the set of previous $t - 1$ responses of this student in CAT.

## 3.1 Expected Gradient Difference Approximation

However, the above selection algorithm is *impractical* in CAT. Because we cannot get student's full responses to bank $Q$, as a result, the gradient difference $\|\nabla l_i(\theta) - \nabla l_j(\theta)\|$ in $w(i, j)$ can not be calculated without related answer correctness labels. Actually, at step $t$, only the responses of previous $t - 1$ steps (i.e., $S_{t-1}$) can be obtained. Therefore, we propose an expected gradient difference approximation method to replace the original to measure their similarity, then the new similarity function $\widetilde{w}(i, j)$ is:

$$\widetilde{w}(i, j) \triangleq d - \max_{\theta \in \Theta} \mathbb{E}_{y \sim p_{\theta^t}} [\|\nabla l_i(\theta) - \nabla l_j(\theta)\|], \tag{7}$$

where the normed gradient difference is calculated as an expectation $\mathbb{E}_{y \sim p_{\theta^t}}$ over the possible labelings, since student's response labels $y$ to the candidate questions are unknown in the selection step. Moreover, for more accurate approximation and to make full use of the available previous $t - 1$ responses, in Eq.(7), the expectation is determined by the current estimate $\theta^t$. This method can be regarded as a gradient difference approximation based on "soft pseudo-labels". Thus, the selection of the next question $q_t$ no longer requires the student's real answer correctness labels:

$$q_t = \arg\max_{q \in Q \setminus S_{t-1}} \Delta(q|S_{t-1}). \tag{8}$$

where $\Delta(q|S_{t-1}) = \widetilde{F}(\{q\} \cup S_{t-1}) - \widetilde{F}(S_{t-1})$, and $\widetilde{F}(S) = \sum_{i \in Q} \max_{j \in S} \widetilde{w}(i, j)$. Also, we uncover some important conclusions about this simple expected approximation:

**Algorithm 1:** The BECAT framework

---

**Require:** $Q$ - question bank, $f$ - IRT or neural network methods.

**Initialize:** Initialize the responses data $S_0 \leftarrow \emptyset$.

1 **for** $t = 1$ **to** $T$ **do**

2      Select question $q_t$ based on $\widetilde{w}(i,j)$:   $q_t \leftarrow \arg\max_{q \in Q \setminus S_{t-1}} \Delta(q|S_{t-1})$.

3      Get student's related answer correctness label $y_t$:   $S_t \leftarrow S_{t-1} \cup \{(q_t, y_t)\}$.

4      Update the weights $\{\gamma_j\}_{j=1}^t$:   $\gamma_j \leftarrow \sum_{i \in Q} \mathbf{1}[j = \arg\max_{s \in S_t} \widetilde{w}(i,s)]$.

5      Update student's ability estimate:   $\theta^t \leftarrow \arg\min_{\theta \in \Theta} \sum_{i \in S_t} \gamma_i l_i(\theta)$.

**Output:** The student's final ability estimate $\theta^T$.

---

**Lemma 1.** *When we replace the original gradient difference in $w(i,j)$ with $\widetilde{w}(i,j)$, the corresponding designed selection algorithm using submodular function $\widetilde{F}$ is actually approximately solving the following optimization problem:*

$$\min_{|S|=T} \max_{\theta \in \Theta} \mathbb{E}_y \left[ \|\sum_{j \in S} \gamma_j \nabla l_j(\theta) - \sum_{i \in Q} \nabla l_i(\theta)\| \right] \tag{9}$$

Based on the conclusion in Lemma 1, we can assume that, after optimization, the preconditioned expected gradient can be approximated by an error of at most $\epsilon$: $\mathbb{E}_y \left[ \|\sum_{i \in S} \gamma_j \nabla l_j(\theta) - \sum_{i \in Q} \nabla l_i(\theta)\| \right] \leq \epsilon$. Then we find the theoretical guarantees for ability estimation when applying gradient-based estimation method to the subset $S$ found by it:

**Theorem 1** (Expected estimation error bound). *Assume that the loss function for ability estimation is $\alpha$-strongly convex (e.g., IRT). Let $S$ be a weighted subset obtained by the proposed method. Then with learning rate $\frac{1}{\alpha}$, ability estimation in gradient descent applied to the subsets has the following expected estimation error bound:*

$$\mathbb{E}\left[\|\theta^{t+1} - \theta^*\|^2\right] \leq \frac{2\epsilon D\alpha + \sigma_l^2 + 2\sigma_f D\alpha H_p(\theta^t, \theta^*)}{\alpha^2} \tag{10}$$

$$where \quad H_p(\theta^t, \theta^*) = \mathbb{E}_{(q,y) \sim p_{\theta^t}} \left[ \frac{1}{p_{\theta^*}(q,y)} \right] \tag{11}$$

*where $\theta^*$ is the optimal estimate using full responses, $\sigma$ is an upper bound on the norm of the gradients, and $D = \max_\theta \|\theta - \theta^*\|$.*

All the proofs can be found in Appendix. The above theorem shows that despite not being able to obtain student's full response, *such simple expected gradient difference approximation can make the estimate error upper bounded at each step*. This theorem is attainable for the case where the loss is strongly convex, such as the cross-entropy loss of the classic L2-regularized IRT [29]. We will further verify the performance of other cases (e.g., neural network-based methods) in experiments.

Theorem 1 also suggests that, to minimize the expected error bound, the CAT systems should try to minimize $H_p(\theta^t, \theta^*)$ that can be regarded as a type of statistical distance: measuring how probability distribution $p_{\theta^t}$ is different from $p_{\theta^*}$. Moreover, we find that with the help of the consistency estimation (i.e., binary cross-entropy) at each step, $H_p(\theta^t, \theta^*)$ can reach its theoretical minimum:

**Theorem 2.** *Assume that $\theta^t$, estimated by the cross-entropy loss in Eq.(1), can minimize the empirical risk i.e., $\sum_{i \in S_t} l_i(\theta^t) = 0$. Then $H_p(\theta, \theta^*)$ can take its minimum when $\theta = \theta^t$, that is*

$$H_p(\theta^t, \theta^*) \leq H_p(\theta, \theta^*), \quad \forall \theta \in \Theta \tag{12}$$

Therefore, the ability estimation methods commonly used in CAT can actually help minimize this upper bound. The proofs and related experiments can be found in the Appendix C and E.4.

**Complexity Analysis of BECAT.**    Algorithm 1 presents the pseudo-code of our BECAT framework. A naive implementation of our selection algorithm in Eq.(8) has the complexity of $O(|Q|^2|\Theta|)$, because at each step we have to: (1) find the question from the bank ($O(|Q|)$) that (2) maximizes

the marginal gain $\Delta(q|S_{t-1}) = \widetilde{F}(\{q\} \cup S_{t-1}) - \widetilde{F}(S_{t-1})$ with complexity $O(|Q||\Theta|)$. To make BECAT faster and more scalable from the above two aspects, we adopt two speed-up tricks: lazy evaluations [30, 31] and multifaceted estimation [32] (See Appendix D for implementation details). Also, we compare the time (second) spent on question selection by different methods in Appendix E.

## 4 Experiments

**Evaluation Method.**    The goal of CAT is to estimate the student's ability accurately with the fewest steps. Therefore, there are usually two tasks to verify the performance of different CAT methods following prior works [9, 12]: **(1) Student Score Prediction**: To evaluate the ability estimate output by CAT, the estimate can be used for predicting the student's binary response (correct/wrong) on the questions he/she has answered in the held-out response data. Thus, Prediction Accuracy (ACC) and AUC are used for evaluations [33]; **(2) Simulation of Ability Estimation**: This is CAT's traditional evaluation methods. Since the ground truth of student ability $\theta_0$ is not available, we artificially generate the $\theta_0$ and further simulate student-question interaction process. Thus, we can use Mean Square Error (MSE) metric. See Appendix E for the details of these two evaluation methods.

**Datasets.**    We conduct experiments on three educational benchmark datasets, namely ASSIST, NIPS-EDU, and EXAM. ASSIST [34] is collected from an online educational system ASSISTments and consists of students' practice logs on mathematics. NIPS-EDU [35] refers to the large-scale dataset in NeurIPS 2020 Education Challenge, which is collected from students' answers to questions from Eedi (an educational platform). The EXAM dataset was supplied by iFLYTEK Co., Ltd., which collected the records of junior high school students on mathematical exams. The statistics of the datasets are shown in appendix. The code can be found in the github: `https://github.com/bigdata-ustc/EduCAT`.

**Compared Approaches.**    To verify the generality of BECAT, in addition to the traditional **IRT**, we also compare the neural network-based model **NeuralCDM** [36]: It can cover many IRT and cognitive diagnosis models, such as MIRT [37] and MF [38, 39]. For the selection algorithm, we mainly use the following SOTA algorithms as baselines: **Random**: The random selection strategy is a benchmark to quantify the improvement of other methods; **FSI** [7] and **KLI** [8] select the question with the maximum Fisher/Kullback-Leibler information, which measures the amount of information that a question carries about the unknown parameter $\theta$. They are specially designed for IRT. **MAAT** [9] utilizes Active Learning [40] to measure the uncertainty caused by each candidate question. **BOBCAT** [12] and **NCAT** [13] recast CAT as a bilevel optimization and Reinforcement Learning problem respectively, and train a data-driven selection algorithm from student response data.

### 4.1   Results and Discussion

In this section, we compare the performance on two classic CAT tasks introduced above to evaluate the effectiveness and efficiency of our proposed BECAT framework. Also, we conduct a qualitative investigation of the characteristics of the selected questions, and gain deeper insights on why BECAT leads to more accurate ability estimation.

**Task1: Student Score Prediction.**    Following prior work [13], we also fix the max length $T = 20$ and calculate the ACC and AUC at step 5, 10 and 20 on three datasets for Student Score Prediction task and the results are shown in Table 3. We find that:

(1) The explicit BECAT framework achieves the best overall performances on the three datasets. It performs significantly better than all the other methods, where the relative performance improvements are as least 1.5% with respect to ACC@20 and 1.1% with respect to AUC@20 on average on ASSIST. This result indicates that BECAT can provide accurate ability estimates at the end of the exam. Also, it even surpasses the implicit selection algorithms based on deep learning, such as NCAT and BOBCAT. This phenomenon shows that *compared to focusing on modeling complex student-question interactions, targeting the accuracy of estimation indeed achieves amazing results.*

(2) BECAT's performance on large-scale datasets (e.g., NIPS-EDU) is better. From Table 3, on NIPS-EDU dataset (the bank size is 27613), BECAT can achieve 2.48% AUC gain (on average) above the famous FSI baseline. On the other two datasets ASSIST and EXAM, the average improvement is only

Table 1: The performance of different methods on Student Score Prediction with ACC and AUC metrics. "–" indicates the information/uncertainty-based selection algorithms (e.g., FSI) cannot be applied to the deep learning method. The boldfaced indicates the statistically significant improvements (p-value < 0.01) over the best baseline.

(a) Performances on ASSIST

| CDM | IRT | | | NeuralCDM | | |
|---|---|---|---|---|---|---|
| Metric@Step | ACC/AUC@5 | ACC/AUC@10 | ACC/AUC@20 | ACC/AUC@5 | ACC/AUC@10 | ACC/AUC@20 |
| Random | 71.01/70.68 | 72.20/71.91 | 73.07/72.61 | 71.52/71.19 | 72.66/72.06 | 72.67/72.83 |
| FSI | 71.77/71.33 | 72.94/72.48 | 73.24/73.54 | – | – | – |
| KLI | 71.93/71.38 | 72.73/72.52 | 73.17/73.57 | – | – | – |
| MAAT | 72.20/71.54 | 72.33/72.58 | 73.22/73.08 | 72.36/70.98 | 72.52/72.33 | 71.74/72.27 |
| BOBCAT | **72.31**/71.68 | 72.36/72.28 | 73.70/73.39 | **72.69**/71.45 | 72.89/72.84 | 73.87/72.84 |
| NCAT | 72.28/71.53 | 72.55/72.31 | 73.81/73.50 | 72.28/71.59 | 72.63/72.37 | 73.90/73.59 |
| **BECAT** | 71.92/71.44 | **73.01/72.73** | **73.96/73.61** | 72.30/**71.60** | **73.11/72.97** | **74.13/73.70** |

(b) Performances on NIPS-EDU

| CDM | IRT | | | NeuralCDM | | |
|---|---|---|---|---|---|---|
| Metric@Step | ACC/AUC@5 | ACC/AUC@10 | ACC/AUC@20 | ACC/AUC@5 | ACC/AUC@10 | ACC/AUC@20 |
| Random | 66.45/69.05 | 68.23/71.66 | 70.23/74.82 | 67.19/69.32 | 68.44/71.56 | 70.57/74.99 |
| FSI | 67.70/70.60 | 69.62/73.62 | 71.03/76.24 | – | – | – |
| KLI | 67.09/69.79 | 69.27/73.30 | 70.42/75.73 | – | – | – |
| MAAT | 66.70/70.32 | 69.13/72.41 | 69.07/74.46 | 67.86/70.12 | 70.07/72.58 | 70.66/75.83 |
| BOBCAT | **69.51/74.42** | 70.94/75.73 | 71.73/76.58 | 71.13/76.00 | 72.52/77.87 | 73.47/79.00 |
| NCAT | 67.30/72.11 | 70.68/75.80 | 71.91/76.66 | 70.47/74.10 | 72.81/77.99 | 73.47/79.12 |
| **BECAT** | 66.98/73.15 | **71.61/75.87** | **72.00/76.82** | **71.33/76.30** | **73.09/78.34** | **73.58/79.36** |

(c) Performances on EXAM

| CDM | IRT | | | NeuralCDM | | |
|---|---|---|---|---|---|---|
| Metric@Step | ACC/AUC@5 | ACC/AUC@10 | ACC/AUC@20 | ACC/AUC@5 | ACC/AUC@10 | ACC/AUC@20 |
| Random | 77.58/70.34 | 78.59/71.91 | 80.40/74.22 | 79.80/72.58 | 79.80/74.81 | 79.80/78.40 |
| FSI | 77.37/70.57 | 78.79/72.21 | 81.01/74.89 | – | – | – |
| KLI | 77.37/70.57 | 78.79/72.21 | 81.01/74.70 | – | – | – |
| MAAT | 76.97/70.38 | 78.79/72.12 | 80.61/74.65 | 82.82/70.32 | 82.83/74.11 | 83.82/79.44 |
| BOBCAT | 80.81/68.17 | 83.84/72.04 | 83.43/72.88 | 78.18/78.24 | 78.19/81.47 | 78.18/79.49 |
| NCAT | 80.92/70.72 | **83.99**/72.71 | 84.02/74.29 | 82.30/**78.77** | 83.19/81.47 | 81.53/79.49 |
| **BECAT** | **80.99/70.74** | 83.85/**72.88** | **84.29/75.00** | 82.84/78.75 | 83.22/81.49 | 84.77/79.70 |

0.32%. This finding inspires us: BECAT is more adaptable to practical large-scale testing situations, and can retrieve the most suitable questions from the massive candidate questions. However, it cannot be ignored that the *BECAT cannot surpass all other methods at the beginning of the exam.* For example, on NIPS-EDU dataset, it is about 2.53% behind BOBCAT on ACC@5. This is because the student's response data available in the initial stage of exam is limited, and the data-driven methods (e.g., BOBCAT and NCAT) can be pre-trained on large-scale student response datasets to learn the interaction patterns, thus addressing this cold-start problem [41]. Thus, adapting the proposed explicit algorithm to data-driven frameworks is a very promising future work.

**Task 2: Simulation of Ability Estimation.** The goal of a practical CAT system is to accurately estimate student's ability. We conduct the Simulation of Ability Estimation experiment on the EXAM dataset using the mean square error $\mathbb{E}[\|\theta^t - \theta_0\|^2]$ between the ability estimate $\theta^t$ and the true ability $\theta_0$ at each step. Figure 3(a) reports the results of different methods on IRT. As the number of questions selected increases, we find that the BECAT method can always achieve much lower estimation errors, especially in the middle stage. Some implicit methods that do not aim at estimation accuracy perform better in the initial stage (e.g., NCAT), but the final accuracy still lags behind BECAT framework. Also, compared with the widely used FSI, the proposed BECAT can reach the same estimation error using up to 20% less questions. On average, it can reach the same estimation accuracy using 15% less questions, which demonstrates its efficiency in ability estimation, i.e., reducing test length.

**The Characteristics of the Selected Questions.** To gain deeper insights on why BECAT leads to more accurate estimation, we take a close look at the characteristics of the selected questions. First, for IRT, we output the difficulty and discrimination parameters of the selected questions and draw a scatter chart in Figure 3(b). We find that it tends to choose those questions with high discrimination,

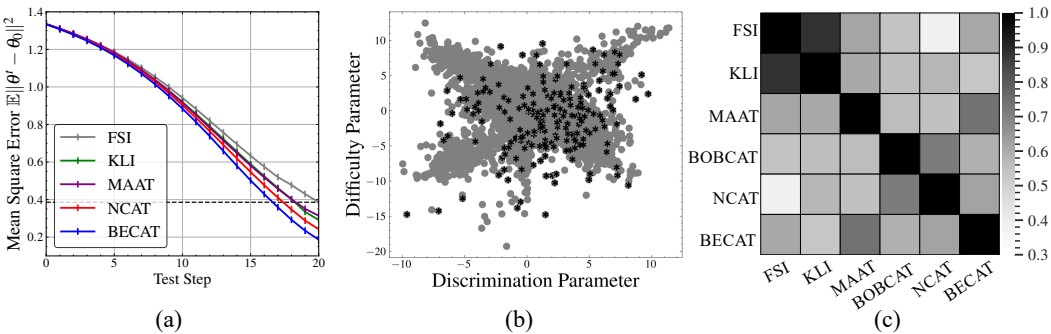

Figure 3: (a) The error of ability estimation on EXAM dataset. (b) The characteristics (i.e., discrimination and difficulty) of the questions selected for 10 students in IRT, where grey dots represent all the questions in the bank, and the "$*$" represent the ones selected by BECAT. (c) The Jaccard similarity coefficient of the selected questions.

and their difficulty is scattered and roughly concentrated in the middle difficulty area, which may be caused by the fact that most of the students are of middle-ability [42]. Second, for NeuralCDM, to gain a better insight into the knowledge concepts (e.g., Geometry in mathematics) covered by the selected questions, and the association between BECAT and other methods. Figure 3(c) shows the Jaccard similarity coefficient of questions' concepts. Questions selected by the same type of method have a high overlap in knowledge concepts, such as FSI and KLI, BOBCAT and NCAT. MAAT and FSI have the highest similarity scores with BECAT: 1) Although BECAT does not directly adopt the concept features in the selection, it has a high score to MAAT that directly targets knowledge concept coverage/diversity, thus making the measurement more comprehensive. 2) The high similarity (with FSI) proves that BECAT is not only general but also capable of selecting informative items.

## 5 Related Works

**Computerized Adaptive Testing**  Computerized Adaptive Testing (CAT) technology has been widely used in many standardized tests, such as GMAT, and the multistage testing in GRE is also its special case of CAT [17]. It is an iterative procedure, mainly including Item Response Theory and a question selection algorithm. The following reviews these two components separately:

*(1) Item Response Theory (IRT).* It is built on psychometric theory and has become popular in educational assessment to provide more individualized feedback about a student's latent ability [43, 44]. It assumes that the examinee's ability is unchanged throughout a test, thus the ability can be estimated using his/her previous response on questions in gradient-based optimization [32]. The classic form is the two-parameter logistic (2PL): $p(\text{the response to question } j \text{ is correct}) = sigmoid(a_j(\theta - b_j))$, where $a_j, b_j \in \mathbb{R}$ represent each question's discrimination and difficulty respectively that are pre-calibrated before testing [29], and $\theta \in \mathbb{R}$ is student's ability to be estimated. Recently, many studies [36, 45, 46] combine cognitive diagnosis and utilize neural networks to model such student-question interaction (e.g., NeuralCDM [36]).

*(2) Selection Algorithms.* The selection algorithm is the core component to realize CAT's adaptivity – accurately estimating student's ability with the fewest test steps. Traditional algorithms are based on some uncertainty or information metrics, e.g., the famous Fisher Information (FSI). Based on it, many methods [8, 47, 48, 49] have been proposed to introduce additional information in selection. Since they are not general and not applicable to recent neural network methods, MAAT [9] uses active learning to select diverse and representative items in question's feature space. Recently, BOBCAT [12] and NCAT [13] regard CAT as a Reinforcement Learning (RL) problem and train selection algorithms directly from large-scale student response data. Due to the unknown of the $\theta_0$, their goal is to minimize the student performance prediction loss of the estimate on the held-out responses data, which is also implicit and prone to biases in training data. In this paper, BECAT is general and explicitly targets the accuracy and efficiency of ability estimation. Compared with previous implicit methods, we find that it exhibits superior performance both theoretically and experimentally. However, various biases, such as those introduced in test item design and respondent pool selection, can affect the validity of estimating a student's true ability [50]. While our approach seeks to improve

the efficiency with which student ability is estimated, it does not diminish the need for test designers to mitigate sources of bias introduced outside of the model fitting process.

**Data Efficiency.** Another closely related literature is data efficiency (or data summary) [51, 15, 52]. To alleviate various costs (computational [53, 54] or labeling costs [40]), data efficiency is used to carefully select or generate some samples from dataset on par with the full data. Its specific implementation methods include Coreset [15, 21, 16, 20], Active Learning [40, 55, 56], Data Distillation [57], etc. For example, recent Coreset approaches [15, 21, 20] try to find a subset that closely approximates the full gradient, i.e., the sum of the gradients of the whole training samples. In this paper, Coreset helps us transform our optimization problem in Section 2.2, but the gradient calculation requires labels, which is obviously not applicable to the CAT scenario (student's response labels cannot be obtained before the question selection). Therefore, we improve it and design an *expected gradient difference approximation* method and provide good upper-bound guarantees to the optimal solution, which is one of the main contributions of this paper.

## 6    Conclusion

This paper focuses on the explicit approach for accurate and efficient estimation of student's true ability $\theta_0$. Given that the ground truth $\theta_0$ is unavailable, we find its theoretical approximation: the ability estimated by the full responses to the question bank, and use it as the optimization goal to design a Bounded Ability Estimation CAT framework (BECAT). For practical use in CAT scenario, we propose a simple but effective expected gradient difference approximation in the greedy selection algorithm. We further analyze its theoretical properties and prove the error upper-bound of the ability estimation on questions found by BECAT. Through extensive experiments on three real-world education datasets, we demonstrate that BECAT can achieve the best estimation accuracy and outperform existing CAT methods at reducing test length.

## Acknowledgments and Disclosure of Funding

This research was partially supported by grants from the National Key Research and Development Program of China (No. 2021YFF0901003), the National Natural Science Foundation of China (Grants No. U20A20229, No. 62106244), and UC Berkeley MicroGrant from the Vice Provost of Undergraduate Education.

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

# A Proofs of Lemma 1

We first proof the following Lemma

**Lemma 1.** *When we replace the original gradient difference in $w(i,j)$ with $\widetilde{w}(i,j)$, the corresponding designed selection algorithm using submodular function $\widetilde{F}$ is actually approximately solving the following optimization problem:*

$$\min_{|S|=T} \max_{\theta \in \Theta} \mathbb{E}_y \left[ \| \sum_{j \in S} \gamma_j \nabla l_j(\theta) - \sum_{i \in Q} \nabla l_i(\theta) \| \right] \tag{13}$$

*Proof.* Following the theoretical analysis in the recent Coreset method CRAIG [15], we first define a mapping function $h$ from set $Q$ to $S$ to a mapping function:$\forall i \in Q, h(i) \in S$. It assign every response data point $i \in Q$ to one of the elements $j$ in $S$. Then, for any arbitrary ability parameter $\theta \in \Theta$ we can write

$$\sum_{i \in Q} \nabla l_i(\theta) = \sum_{i \in Q} [\nabla l_i(\theta) - \nabla l_{h(i)}(\theta) + \nabla l_{h(i)}(\theta)] \tag{14}$$

$$= \sum_{i \in Q} [\nabla l_i(\theta) - \nabla l_{h(i)}(\theta)] + \sum_{j \in S} \gamma_j \nabla l_j(\theta) \tag{15}$$

Subtracting and taking the expected norm of the both sides, we get an upper bound on the error. According to the triangle inequality, we have

$$\mathbb{E} \left[ \| \sum_{i \in Q} \nabla l_i(\theta) - \sum_{i \in S} \gamma_j \nabla l_j(\theta) \| \right] \le \sum_{i \in Q} \mathbb{E} \left[ \| \nabla l_i(\theta) - \nabla l_{h'(i)}(\theta) \| \right]. \tag{16}$$

When the mapping function $h$ is to map each element in $Q$ to the one in $S$ that is closest to its expected gradient, the right side of inequality (16) is minimized, or minimum expected distance between the gradient: $h(i) = \arg\min_{j \in S} \mathbb{E} [\| \nabla l_i(\theta) - \nabla l_j(\theta) \|]$. Therefore, the upper bound of the expected gradient difference can be further constrained:

$$\min_{|S|=T} \mathbb{E} \left[ \| \sum_{i \in Q} \nabla l_i(\theta) - \sum_{i \in S} \gamma_j \nabla l_j(\theta) \| \right] \le \sum_{i \in Q} \min_{j \in S} \mathbb{E} [\| \nabla l_i(\theta) - \nabla l_j(\theta) \|]. \tag{17}$$

Next, define a similarity function $\widetilde{w}(i,j)$ which measures the expected gradient similarity between response pair $i$ and $j$: $\widetilde{w}(i,j) = d - \max_{\theta \in \Theta} \mathbb{E} [\| \nabla l_i(\theta) - \nabla l_j(\theta) \|]$, and $d = \max_{i \in Q, j \in S} \max_{\theta \in \Theta} \| \nabla l_i(\theta) - \nabla l_j(\theta) \|$ is the maximum pairwise gradient distance. Thus, the optimization problem (Eq.(13)) can also be transformed as:

$$\max_{|S|=T} \sum_{i \in Q} \max_{j \in S} \widetilde{w}(i,j). \tag{18}$$

Following the same way of origin problem, its corresponding submodular $\widetilde{F}(S) = \sum_{i \in Q} \max_{j \in S} \widetilde{w}(i,j)$, which is the same with our proposed method. Thus, the designed selection algorithm is the greedy algorithm of the optimization problem (Eq.(13)).

$\square$

# B Proofs of Theorem 1

**Theorem 1** (Expected estimation error bound)**.** *Assume that the loss function for ability estimation is $\alpha$-strongly convex (e.g., IRT). Let $S$ be a weighted subset obtained by the proposed method. Then with learning rate $\frac{1}{\alpha}$, ability estimation in gradient descent applied to the subsets has the following expected estimation error bound:*

$$\mathbb{E} \left[ \| \theta^{t+1} - \theta^* \|^2 \right] \le \frac{2\epsilon D\alpha + \sigma_l^2 + 2\sigma_f D\alpha H_p(\theta^t, \theta^*)}{\alpha^2} \tag{19}$$

$$where \quad H_p(\theta^t, \theta^*) = \mathbb{E}_{(q,y) \sim p_{\theta^t}} \left[ \frac{1}{p_{\theta^*}(q,y)} \right] \tag{20}$$

*where $\theta^*$ is the optimal estimate using full responses, $\sigma$ is an upper bound on the norm of the gradients, and $D = \max_\theta \|\theta - \theta^*\|$.*

*Proof.* We now provide the expected estimation error bound for strongly convex functions building on the analysis of [15, 58]. Let $g^t = \frac{1}{|Q|} \sum_{i \in Q} \nabla l_i(\theta^t)$, $g_S^t = \sum_{i \in S} \gamma_i \nabla l_i(\theta^t)$, and normalize the subset weights at every iteration i.e., $\sum_{j \in S} \gamma_j = 1$. Let $L(\theta) = \sum_{i \in S} \gamma_i l_i(\theta)$ be the weighted subset training loss parameterized by ability parameters $\theta$, and we have:

$$
\begin{aligned}
\|\theta^{t+1} - \theta^*\|^2 &= \|\theta^t - \eta g_S^t - \theta^*\|^2 \\
&= \|\theta^t - \theta^*\|^2 - 2\eta (g_S^t)^\top (\theta^t - \theta^*) + \eta^2 \|g_S^t\|^2 \\
&\leq \|\theta^t - \theta^*\|^2 - 2\eta [L(\theta^t) - L(\theta^*)] + \eta^2 \|g_S^t\|^2 \\
&\leq \|\theta^t - \theta^*\|^2 - 2\eta \left[ (g_S^*)^\top (\theta^t - \theta^*) + \frac{\alpha}{2} \|\theta^t - \theta^*\|^2 \right] \\
&\quad + \eta^2 \|g_S^t\|^2 \quad (\alpha - strongly\ convex).
\end{aligned}
\tag{21}
$$

According to Cauchy–Schwarz inequality, we have

$$
|(g_S^*)^\top (\theta^t - \theta^*)| \leq \|g_S^*\| \|\theta^t - \theta^*\|.
\tag{22}
$$

Thus

$$
\|\theta^{t+1} - \theta^*\|^2 \leq \|\theta^t - \theta^*\|^2 + 2\eta \left[ \|g_S^*\| \|\theta^t - \theta^*\| \right] - \eta\alpha \|\theta^t - \theta^*\|^2 + \eta^2 \|g_S^t\|^2
\tag{23}
$$

Taking expectation with respect to the randomness in the label (i.e., the correctness of response) decided by $\theta^t$, we have

$$
\begin{aligned}
\mathbb{E}\left[\|\theta^{t+1} - \theta^*\|^2\right] &\leq (1 - \eta\alpha)\mathbb{E}\left[\|\theta^t - \theta^*\|^2\right] \\
&\quad + 2\eta\mathbb{E}\left[\|g_S^*\| \|\theta^t - \theta^*\|\right] + \eta^2 \mathbb{E}\left[\|g_S^t\|^2\right].
\end{aligned}
\tag{24}
$$

Assuming gradients have a bounded norm $\|\nabla l_j(\theta)\| \leq \sigma_l$ and $\|\nabla f_j(\theta)\| \leq \sigma_f$. Thus, from reverse triangle inequality, we can write

$$
\mathbb{E}\left[\|g_S^t\|^2\right] = \mathbb{E}\left[\|\sum_{j \in S} \gamma_j \nabla l_j(\theta^t)\|^2\right] \leq \sigma_l^2.
\tag{25}
$$

From Lemma 1, we can assume that the subset $S$ and corresponding per-element weights $\gamma_j$ can approximate the full gradient with an expected error at most $\epsilon > 0$, i.e., $\mathbb{E}_{y \sim p_{\theta^t}}\left[\|\sum_{i \in Q} \nabla l_i(\theta) - \sum_{i \in S} \gamma_j \nabla l_j(\theta)\|\right] \leq \epsilon$. Thus, from reverse triangle inequality $\mathbb{E}[\|g_S^*\|] \leq \mathbb{E}[\|g^*\|] + \epsilon$ and $\mathbb{E}[\|g^*\|]$ can be further derived as follows:

$$
\begin{aligned}
\mathbb{E}[\|g^*\|] &= \frac{1}{|Q|}\mathbb{E}\left[\|\sum_{i \in Q} \nabla l_i(\theta^*)\|\right] \leq \frac{1}{|Q|}\sum_{i \in Q}\mathbb{E}\left[\|\nabla l_i(\theta^*)\|\right] \\
&= \frac{1}{|Q|}\sum_{i \in Q}\mathbb{E}_{y_i \sim p_{\theta^t}}\left[\|\nabla_{\theta = \theta^*}(-y_i \ln f_i(\theta) - (1-y)\ln(1 - f_i(\theta)))\|\right] \\
&= \frac{1}{|Q|}\sum_{i \in Q}\mathbb{E}_{y_i \sim p_{\theta^t}}\left[\|-\frac{y_i}{f_i(\theta^*)}\nabla f_i(\theta^*) + \frac{1 - y_i}{1 - f_i(\theta^*)}\nabla f_i(\theta^*)\|\right] \\
&= \frac{1}{|Q|}\sum_{i \in Q}\|\nabla f_i(\theta^*)\|\mathbb{E}_{y_i}\left|\frac{y_i}{f_i(\theta^*)} - \frac{1 - y_i}{1 - f_i(\theta^*)}\right| \\
&= \frac{1}{|Q|}\sum_{i \in Q}\|\nabla f_i(\theta^*)\|\sum_{y \in \{0,1\}}\frac{p_{\theta^t}(q_i, y_i = y)}{p_{\theta^*}(q_i, y_i = y)} \\
&= \frac{1}{|Q|}\sum_{i \in Q}\|\nabla f_i(\theta^*)\|\mathbb{E}_{(q_i, y_i) \sim p_{\theta^t}}\left[\frac{1}{p_{\theta^*}(q_i, y_i)}\right] \\
&\leq \sigma_f \mathbb{E}_{(q,y) \sim p_{\theta^t}}\left[\frac{1}{p_{\theta^*}(q, y)}\right],
\end{aligned}
\tag{26}
$$

where $p_\theta(q, y)$ is the response distribution of the student with the ability $\theta$, and $f_i(\theta) = p_\theta(q_i, y_i = 1)$ is the output of IRT. Also assuming that $\|\theta - \theta^*\| \le D$, we have,

$$\mathbb{E}\left[\|g_S^*\|\|\theta^t - \theta^*\|\right] \le D\mathbb{E}[\|g^*\|] + \epsilon D$$

$$\le \sigma_f D\mathbb{E}_{(q,y)\sim p_{\theta^t}}\left[\frac{1}{p_{\theta^*}(q, y)}\right] + \epsilon D. \tag{27}$$

Combining the Equation (24) to (26) and let $\eta = \frac{1}{\alpha}$, we have

$$\mathbb{E}\left[\|\theta^{t+1} - \theta^*\|^2\right] \le \frac{2\epsilon D\alpha + \sigma_l^2}{\alpha^2} + \frac{2\sigma_f D}{\alpha}\mathbb{E}_{(q,y)\sim p_{\theta^t}}\left[\frac{1}{p_{\theta^*}(q, y)}\right]$$

$$= \frac{2\epsilon D\alpha + \sigma_l^2 + 2\sigma_f D\alpha H_p(\theta^t, \theta^*)}{\alpha^2}, \tag{28}$$

where $H_p(\theta^t, \theta^*) = \mathbb{E}_{(q,y)\sim p_{\theta^t}}\left[\frac{1}{p_{\theta^*}(q, y)}\right]$. So we complete the proof.

$\square$

## C Proofs of Theorem 2

**Theorem 2.** *Assume that $\theta^t$, estimated by the cross-entropy loss, can minimize the empirical risk i.e., $\sum_{i\in S_t} l_i(\theta^t) = 0$. Then $H_p(\theta, \theta^*)$ can take its minimum when $\theta = \theta^t$, that is*

$$H_p(\theta^t, \theta^*) \le H_p(\theta, \theta^*), \quad \forall \theta \in \Theta \tag{29}$$

*Proof.* When we use the binary cross-entropy (BCE) loss to estimate the ability $\theta$ and minimize the empirical risk at step $t$, we have

$$\theta^t = \arg\min_\theta \sum_{i\in S_t} l_i(\theta)$$

$$= \arg\max_\theta \sum_{i\in S_t} y_i \log f_i(\theta) + (1 - y_i)\log(1 - f_i(\theta))$$

$$= \arg\max_\theta \sum_{i\in S_t} \log p_\theta(q_i, y_i)$$

$$\approx \arg\max_\theta \mathbb{E}_{(q,y)\sim p_{\theta_0}} \log p_\theta(q, y), \tag{30}$$

where $f_i(\theta) = p_\theta(q_i, y_i = 1)$ is the output of the IRT. We argue that the student's response to the question $q$ is determined by the true ability $\theta_0$ in the CAT process, i.e., $y_i = \arg\max_y p_{\theta_0}(q_i, y)$. Therefore, when the above empirical risk achieves its minimum, i.e., $\sum_{i\in S_t} l_i(\theta^t) = 0$, $\mathbb{E}_{(q,y)\sim p_{\theta_0}} \log p_\theta(q_i, y_i)$ can reach the maximum 0, and the real and predicted responses are the same:

$$y_{max} = \arg\max_y p_{\theta^t}(q, y) = \arg\max_y p_{\theta_0}(q, y), \forall q \tag{31}$$

and the predicted probability should approache 1: $p_{\theta^t}(q, y_{max}) \to 1$.

Next, we discuss the minimization of the statistical distance $H_p(\theta^t, \theta^*) = \mathbb{E}_{(q,y)\sim p_{\theta^t}}\left[1/p_{\theta^*}(q, y)\right]$ term in the upper bound in Theorem 1. In general, make now the simplifying assumption that distribution $p_{\theta^*}(q, y)$ is smooth and has a global maximum probability $p_{max}$, attained at a point $(q, y_{max})$ for question $q$, so that $1/p_{\theta^*}(q, y)$ has a global minimum at the same $y_{max}$. Obviously, we can choose:

$$p_{\theta^t}(q, y) = \delta(y - y_{max}), \tag{32}$$

where $\delta$ is the delta function [59], making $\mathbb{E}_{(q,y)\sim p_{\theta^t}}\left[1/p_{\theta^*}(q, y)\right]$ to its minimum. Based on the above findings, in the CAT situation (binary classification), the optimal distribution $p_{\theta^t}$ (minimizing the $H_p$ in the upper bound) needs to: have the same classification result as $p_{\theta^*}$, i.e., $y_{max} = \arg\max_y p_{\theta^t}(q, y) = \arg\max_y p_{\theta^*}(q, y) \approx \arg\max_y p_{\theta_0}(q, y)$, and its corresponding probability is as large as possible, i.e., $p_{\theta^t}(q, y_{max}) \to 1$. In this case, $H_p(\theta^t, \theta^*)$ can takes its minimum. All of these findings are consistent with the conclusion that minimizing BCE loss for ability estimation in Eq.(30) and (31). So we complete the proof.

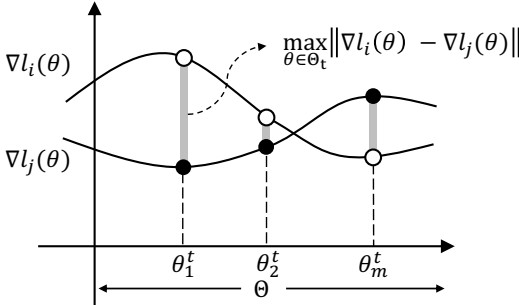

Figure 4: The illustration of the distance measurements in similarity function $\widetilde{w}(i,j)$: replace the entire ability space $\Theta$ with the student's possible ability estimates $\Theta_t = \{\theta_i^t\}_{i=1}^m$ to reduce search space and speed up the selection.

## D   Implementation Details of BECAT

To improve its complexity, we provide two implementation tricks:

**(1) Lazy Evaluations.**   By exploiting the submodularity, we use the lazy evaluations approach presented in [30, 31] to speed up the selection process and make the running time faster in practice. At step $t$, the greedy selection algorithm must identify the question $q$ with maximum marginal gain $\Delta(q|S_{t-1})$. Instead, this lazy method involves using a max-heap ($O(1)$ lookup and $O(\log(n))$ insertion) to keep an upper bound on the gain of each question that comes from submodularity, i.e., the marginal benefits of any question $q \in Q$ are monotonically nonincreasing during the selection:

$$\Delta(q|S_{t-1}) \geq \Delta(q|S_t) \quad \forall q \in Q. \tag{33}$$

Instead of recomputing $\Delta(q|S_{t-1})$ at each step for each element $q \in Q$ (requiring $O(|Q|)$ computations), the accelerated lazy algorithm maintains a list of upper bounds $\Delta'(q)$ (initialized to $\infty$) on the marginal gains sorted in decreasing order (max-heap order). Specifically, at each step, the algorithm first selects the maximal from this ordered list, i.e., the top of the heap. It then updates this bound $\Delta'(q) \leftarrow \Delta(q|S_{t-1})$ in the heap. As soon as the $\Delta'(q)$ is still at the top of the heap, then submodularity Eq.(33) guarantees that $\Delta(q|S_{t-1}) \geq \Delta(q'|S_{t-1})$ for all $q' \neq q$, and therefore we do not need to evaluate any more items. If it does not satisfy this condition, we just insert it with $\Delta'(q)$ as the new upper bound and repeat the above procedure until the qualified question is selected. While the worst case is the same, in practice this method has enormous speedups over the standard greedy algorithm [60].

**(2) Reducing the Ability Space $\Theta$**   In our method, note that the similarity function $\widetilde{w}(i,j) = d - \max_{\theta \in \Theta} \mathbb{E}_{y \sim p_{\theta^t}} [\|\nabla l_i(\theta) - \nabla l_j(\theta)\|]$ requires finding the worst case over the entire ability parameter space $\Theta$, which is too expensive for CAT systems. In fact, it only needs to be calculated in each student's own possible ability space. In other words, we can find an ability subspace $\Theta_t \subseteq \Theta$ specialized to each student. We utilize a novel MLE-based estimation approach [32], which models student's multifaceted nature and sequentially generates a set of possible abilities $\Theta_t = \{\theta_i^t\}_{i=1}^m$ at each step $t$:

$$\theta_i^t = \arg\min_{\theta_i} \sum_{i \in S_t} l_i(\theta_i) - \frac{\lambda}{2} \left\|\theta_i - \bar{\theta}_i\right\|^2 \quad for\ i = 1, ..., m, \tag{34}$$

where $\bar{\theta}_i$ is the average of previous $i - 1$ estimates and the term $\left\|\theta_i - \bar{\theta}_i\right\|^2$ ensures the diversity of abilities in $\{\theta_i^t\}_{i=1}^m$. We refer the reader to [32] for more details about this estimation method. As shown in Figure 4 we replace the entire parameter space in the optimization problem with student's $m$ potential estimates $\Theta_t = \{\theta_i^t\}_{i=1}^m$. The complexity can be reduced to $O(|Q|^2 m)$ and $m \ll |\Theta|$.   $\square$

## E   Details of Experiment

**The impact of test length.**   We use simulation experiments to verify the choice of max length $T$: Due to the unknown of the true ability $\theta_0$, we artificially generate it and conduct the Simulation

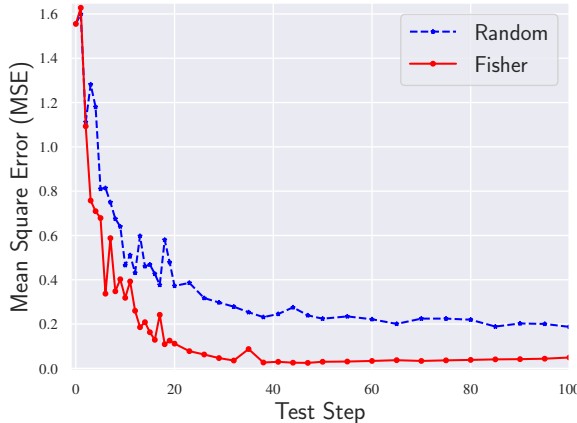

Figure 5: Simulation experiments of ability estimation using MSE: $\mathbb{E}[\|\theta^t - \theta_0\|^2]$.

of Ability Estimation experiment on the EXAM dataset using the mean square error $\mathbb{E}[\|\theta^t - \theta_0\|^2]$ between the ability estimate $\theta^t$ at each step and the true ability $\theta_0$ (Figure 5): The classic Fisher method can reduce the evaluation error quickly and 20 is sufficient for the length of a typical adaptive test. Thus, we fix the max length $T = 20$.

**Experimental Implementation Details.** As 20 is sufficient for the length of a typical test [13], we also fix the max length $T = 20$. We implement all the methods with PyTorch. We set batch size to 64 and the learning rate to 0.001, and optimize all the parameters using the Adam algorithm [61] on a Tesla V100-SXM2-32GB GPU.

## E.1 Statistics of the datasets

Table 2: Statistics of the datasets

| Dataset | ASSIST | NIPS-EDU | EXAM |
|---|---|---|---|
| #Students | 20,704 | 220,274 | 9,214 |
| #Questions | 15,071 | 27,613 | 1,650 |
| #Response logs | 1,768,253 | 19,181,192 | 133,398 |
| #Response logs per student | 85.41 | 87.08 | 14.48 |
| #Response logs per question | 117.33 | 694.64 | 80.85 |

## E.2 Detailed Evaluation Method

The goal of CAT is to estimate the student's ability accurately with the fewest steps. However, since the true ability cannot be obtained as the ground truth, there are usually two tasks to verify the performance of different CAT methods following previous works [9, 12]: 1) Student Score Prediction and 2) Simulation of Ability Estimation:

*1) Student Score Prediction.* To evaluate the ability estimate output by the CAT system, this estimate can be used for predicting the student's scores (correct or wrong) on the questions he/she has answered in the held-out response data. This is an indirect evaluation method. Following the common strategy [12], we use 70%-20%-10% students for training, validation, and testing respectively, and the students in validation/testing set won't appear in training. The training set is used for initializing some question's parameters in IRT (e.g., difficulty parameter in IRT), and the data-driven selection algorithm baselines (e.g., BOBCAT). In the validation/testing, the responses of each student $i$ are further divided into the candidate ($Q_i$) and meta ($M_i$) question sets to simulate CAT procedure, following [9, 12]. Specifically, at each step, (1) different selection algorithms first select a question from $Q_i$; (2) IRT then updates the ability estimate with his/her response to it; (3) evaluate this estimate's accuracy by predicting binary-valued responses (correct or wrong) on the held-out meta set $M_i$. This task assumes that the more accurate the score prediction is, the more accurate the ability

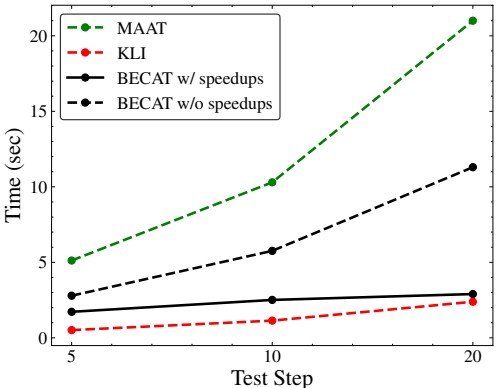

Figure 6: Comparison of the time consumed by different methods at the selection step. The implementation tricks provide $2\times$ to $3\times$ speedup.

estimate is. This task considers that the accuracy of score prediction can reflect the accuracy this ability estimates. Thus, from this binary classification perspective, we use Prediction Accuracy (ACC) [46] and Area Under ROC Curve (AUC) [62] for the evaluation of different selection algorithms.

*2) Simulation of Ability Estimation.* This is CAT's traditional evaluation method [2]. Since the ground truth of student ability $\theta_0$ is not available, we *artificially generate* their $\theta_0$ and further simulate student-question interaction process within CAT systems. For the rationality of the generated $\theta_0$, we use all the students' responses in the dataset to estimate their ability $\{\theta_0^1, \theta_0^2, ..., \theta_0^N\}$ as the ground truth [9, 63]. Also, the dataset is used to learn questions' parameters and fix them. Different from the first task, such settings can simulate students with $\theta_0$ responding to any question in $Q$, thus the candidate question in selection is the entire bank $Q$. Specifically, (1) different selection algorithms first select a question from the entire bank $Q$; (2) IRT then updates the estimate with the response to it; (3) evaluate this estimate's accuracy by computing the difference between the estimated and the true ability. In this way, we can use Mean Square Error (MSE) to evaluate the accuracy of estimation.

### E.3 Implementation Tricks for Speedups

To solve the BECAT optimization problem, we need to calculate the gradients of all items in question bank, leading to high computation requirements for large datasets/examinations. To this end, we consider two speed-up tricks: lazy evaluations and multifaceted estimation. Lazy evaluations take advantage of submodularity to avoid calculating the conditional gain $\Delta(q|S_{t-1})$ of all the candidate items. The multifaceted estimation method [32] can effectively reduce the ability space when calculating the similarity between two items, thus reducing the time to calculate $\Delta(q|S_{t-1})$ of each item. In Figure 6, we compare the time (second) spent on question selection by different methods[3], and find that the proposed implement tricks give the average speedup of $3\times$, and achieve $7\times$ to recent MAAT. And it is almost the same time as the traditional informativeness-based method KLI. This demonstrates that our greedy selection algorithm in BECAT is fast in practice.

### E.4 BECAT Analysis

In this section, we will further analyze its selection time/latency, the effectiveness of Theorem 1 and 2, and the characteristics of questions selected by BECAT, respectively.

**Upper Bound Analysis of Estimation** In Theorem 1, the most important component is $H_p(\theta^t, \theta^*)$, which determines the upper bound on estimation error and the convergence behavior of the proposed BECAT method. Therefore, to verify the effectiveness and theoretical guarantees of our explicit selection algorithm, i.e., the estimation error bound conclusion in Theorem 1, we compute $H_p$ for each step on the strongly convex loss function: the cross-entropy loss of the L2-regularized IRT. This experiment is still based on the simulation setting in Appendix E.2 and the results are shown in

---

[3]There is no comparison here of reinforcement learning methods, due to the extra training overhead they require.

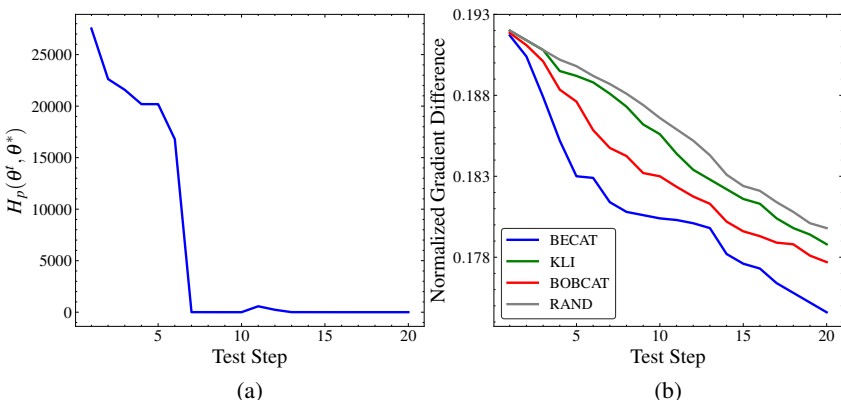

(a)                                      (b)

Figure 7: (a) The value of $H_p(\theta^t, \theta^*)$ in the upper-bound of estimation. (b) Normed difference between the full gradient on $|Q|$ and the gradient of the question subset found by different methods.

Figure 7(a): In the first few steps, the $H_p(\theta^t, \theta^*)$ can rapidly decrease to the minimum. As the test progresses, the ability estimate tends to be accurate (Figure 7(a)), and the upper-bound $H_p$ remains near the minimum, which reflects the good convergence behavior of BECAT. This demonstrates that our expected gradient difference approximation is reliable in practice.

**Gradient Approximation.** In Proposition 1 we show that $\theta^*$ is an approximation of student true ability. To approach the new target $\theta^*$, an approximation to the full gradient of the bank $Q$ is required. Figure 7(b) demonstrates the norm of the difference between the weighted gradient of the questions subset found by BECAT and the full gradient. This figure also compares the normed gradient difference of other subsets found by other methods where each response is weighted by $|Q|/|S|$. The gradient difference is calculated by sampling the full gradient at various points in the parameter space. Note that the gradient difference obtained by BECAT decreases significantly with the increase of $t$ and it is much smaller than that of other methods, which proves that the expected gradient difference approximation method is accurate. Combining the experimental results on the two tasks in Experiments, the better the prediction/estimation performance of the method, the smaller the gradient difference. This demonstrates that the closeness to $\theta^*$ reflects the closeness to the true ability $\theta_0$, which further proves the rationality of using $\theta^*$ as a new target.

Table 3: The variance results of different methods on ACC and AUC metrics.

(a) Variance results on ASSIST

| CDM | IRT | | | NeuralCDM | | |
|---|---|---|---|---|---|---|
| Metric@Step | ACC / AUC@5 | ACC / AUC@10 | ACC / AUC@20 | ACC / AUC@5 | ACC / AUC@10 | ACC / AUC@20 |
| Random | 0.0378 / 0.0462 | 0.0090 / 0.0226 | 0.0052 / 0.0142 | 0.0064 / 0.0072 | 0.0013 / 0.0042 | 0.0012 / 0.0038 |
| FSI | 0.0446 / 0.0257 | 0.0147 / 0.0076 | 0.0067 / 0.0024 | – | – | – |
| KLI | 0.0163 / 0.0058 | 0.0042 / 0.0037 | 0.0023 / 0.0019 | – | – | – |
| MAAT | 0.0121 / 0.0202 | 0.0082 / 0.0150 | 0.0162 / 0.0286 | 0.0123 / 0.0073 | 0.0083 / 0.0276 | 0.0253 / 0.0242 |
| BOBCAT | 0.0120 / 0.0152 | 0.0052 / 0.0057 | 0.0165 / 0.0148 | 0.0054 / 0.0066 | 0.0047 / 0.0024 | 0.0126 / 0.0023 |
| NCAT | 0.0063 / 0.0041 | 0.0065 / 0.0055 | 0.0012 / 0.0007 | 0.0032 / 0.0031 | 0.0037 / 0.0029 | 0.0035 / 0.0020 |
| **BECAT** | 0.0100 / 0.0136 | 0.0062 / 0.0040 | 0.0055 / 0.0023 | 0.0022 / 0.0022 | 0.0019 / 0.0011 | 0.0012 / 0.0010 |

(b) Variance results on NIPS-EDU

| CDM | IRT | | | NeuralCDM | | |
|---|---|---|---|---|---|---|
| Metric@Step | ACC / AUC@5 | ACC / AUC@10 | ACC / AUC@20 | ACC / AUC@5 | ACC / AUC@10 | ACC / AUC@20 |
| Random | 0.0185 / 0.0406 | 0.0183 / 0.0431 | 0.0182 / 0.0478 | 0.0185 / 0.0406 | 0.0183 / 0.0431 | 0.0182 / 0.0478 |
| FSI | 0.0270 / 0.0350 | 0.0269 / 0.0357 | 0.0274 / 0.0401 | – | – | – |
| KLI | 0.0231 / 0.0315 | 0.0218 / 0.0278 | 0.0196 / 0.0247 | – | – | – |
| MAAT | 0.0207 / 0.0347 | 0.0232 / 0.0377 | 0.0256 / 0.0412 | 0.0192 / 0.0298 | 0.0216 / 0.0306 | 0.0253 / 0.0363 |
| BOBCAT | 0.0203 / 0.0311 | 0.0185 / 0.0267 | 0.0169 / 0.0247 | 0.0197 / 0.0315 | 0.0190 / 0.0289 | 0.0200 / 0.0310 |
| NCAT | 0.0178 / 0.0246 | 0.0159 / 0.0214 | 0.0142 / 0.0198 | 0.0176 / 0.0258 | 0.0163 / 0.0232 | 0.0169 / 0.0246 |
| **BECAT** | 0.0216 / 0.0294 | 0.0185 / 0.0248 | 0.0169 / 0.0225 | 0.0204 / 0.0304 | 0.0167 / 0.0251 | 0.0165 / 0.0225 |

(c) Variance results on EXAM

| CDM | IRT | | | NeuralCDM | | |
|---|---|---|---|---|---|---|
| Metric@Step | ACC / AUC@5 | ACC / AUC@10 | ACC / AUC@20 | ACC / AUC@5 | ACC / AUC@10 | ACC / AUC@20 |
| Random | 0.0171 / 0.0189 | 0.0114 / 0.0136 | 0.0084 / 0.0086 | 0.0025 / 0.0107 | 0.0071 / 0.0102 | 0.0070 / 0.0090 |
| FSI | 0.0195 / 0.0112 | 0.0097 / 0.0099 | 0.0093 / 0.0069 | – | – | – |
| KLI | 0.0195 / 0.0112 | 0.0097 / 0.0099 | 0.0076 / 0.0098 | – | – | – |
| MAAT | 0.0193 / 0.0115 | 0.0104 / 0.0094 | 0.0082 / 0.0065 | 0.0191 / 0.0112 | 0.0182 / 0.0099 | 0.0053 / 0.0062 |
| BOBCAT | 0.0036 / 0.0133 | 0.0070 / 0.0114 | 0.0086 / 0.0092 | 0.0142 / 0.0087 | 0.0095 / 0.0091 | 0.0012 / 0.0021 |
| NCAT | 0.0033 / 0.0146 | 0.0005 / 0.0004 | 0.0004 / 0.0004 | 0.0160 / 0.0120 | 0.0148 / 0.0090 | 0.0003 / 0.0003 |
| **BECAT** | 0.0196 / 0.0121 | 0.0022 / 0.0006 | 0.0020 / 0.0007 | 0.0172 / 0.0134 | 0.0063 / 0.0012 | 0.0090 / 0.0029 |

