# OpenReview forum: "A Bounded Ability Estimation for Computerized Adaptive Testing"
_NeurIPS.cc/2023/Conference — NeurIPS 2023 poster_

### Official Review · Reviewer_DatR · 2023-06-27

**Soundness:** 2 fair
**Presentation:** 3 good
**Contribution:** 3 good
**Rating:** 5
**Confidence:** 3

**Summary:**

This is a very interesting paper that proposes a coreset-based algorithm to evaluate students' ability to answer questions from a question bank. The main innovations are defining a true student's probability and proposing a coreset algorithms with pseudo labels.

**Strengths:**

**originality**\
I didn't have direct experience in CAT; however, the application of the coreset finding in efficient ability estimation for CAT is promising.

**quality**\
The proposed algorithm logically makes sense to me, and the English writing is generally clear enough.

**clarity**\
The organization of the paper is good, which helps me understand the main content quickly.

**significance**\
I am a bit concerned about the ethical side of the work. The corset finding is dependent on the test takers, which means different test takers would end up with different questions assigned. Although the approximation error to the true estimation is claimed and given, I think it may lead to the fairness concern of whether giving different questions to different test takers is fair.


**Weaknesses:**

I have multiple concerns regarding the mathematic rigor in this paper.

1. In proposition 1, the definition of question bank $Q$ is missing. Hence, how "questions" or samples are generated to increase $|Q|$ is unclear. In addition, I think the proof of Proposition 1 should add $\forall \epsilon>0$.

2. Definition 2 seems correct. However, I think it is nontrivial to prove that optimizing a loss function with gradients equivalent to the original loss function is equivalent to optimizing the original loss function.

3. The upper bound for (4) is skipped in line 148. It would be clearer to list the upper bound of the loss first and then give the solution (coreset) of this upper bound.

4. lemma 1 actually does not say much useful for the approximation. If you want to say the algorithm is to approximately solve the problem, then an approximation error to measure the distance between the original function and the function approximation should be explicitly provided.

5. The expectation writing in line 188 is problematic. What distribution does $Y$ come from?

6. Why the ''probability distance'' $H_p(\theta^t, \theta^*)$ in (10) is not zero when $\theta^t=\theta^*$?



**Questions:**

Can you provide the variance of your experimental results?

**Limitations:**

Please see the weakness section.

---

> ### Author Rebuttal · Authors · 2023-08-07
>
> Thank you for your valuable feedback on our CAT paper. We appreciate the time and effort you have invested in reviewing our work.
>
> For your concerns regarding the fairness of CAT itself, first of all, let me introduce the background of CAT: as stated in Section 1, CAT is a *personalized* question selection system, which can adaptively select suitable items for individuals, assessing their ability accurately and efficiently. The typical application of CAT is GRE exam: when a test-taker begins the GRE, the computer provides a question of medium difficulty. If the response is correct, the computer presents a slightly more challenging question. Conversely, if the response is incorrect, the following question will be slightly easier. That's why some candidates may perceive the test becoming progressively more difficult.
>
> Therefore, similar to the recommendation system (different users will be recommended different items), *CAT also has fairness issues of course*. But this is another interesting topic, which is not the focus of this paper. Following your and Ethics Reviewer xQDC's suggestion, in an improved version, we will give a brief overview of fairness issues in CAT itself. The following are the responses to your other questions:
>
> > **Q1**: In proposition 1, the definition of question bank $Q$ is missing. Hence, how "questions" or samples are generated to increase $|Q|$ is unclear. In addition, I think the proof of Proposition 1 should add $\forall \epsilon >0$.
>
> **A1**: The question bank $Q$ is defined in the beginning of Section 2. Proposition 1 tries to explain that as the bank size $|Q|$ increases, the approximation becomes more ideal, and increasing the question bank is obviously achievable in real scenarios. Regarding $\epsilon$, generally in the mathematical theory of CAT, $\epsilon$ is greater than 0 by default. Sorry to confuse you, we will make the derivation more detailed in the follow-up.
>
> > **Q2**: Definition 2 seems correct. However, I think it is nontrivial to prove that optimizing a loss function with gradients equivalent to the original loss function is equivalent to optimizing the original loss function.
>
> **A2**: The transformation in Definition 2 is intuitive and consistent with Coreset method: in an optimization problem, if you use different dataset for standard gradient descent, as long as their gradient sum ($\sum_{j\in S}{\gamma_j \nabla l_j(\theta)}$ and $\sum_{i\in Q}{\nabla l_i(\theta)}$) and initial parameter point are guaranteed to be the same, then the final optimization result (${\theta^T}$ and $\theta^*$) will naturally be the same. If you have any other questions, please feel free to ask.
>
> > **Q3**: The upper bound for (4) is skipped in line 148. It would be clearer to list the upper bound of the loss first and then give the solution (coreset) of this upper bound.
>
> **A3**: Since this part is based on previous Coreset research (cited in the paper), and considering the space limit, this part of the derivation is skipped. In the subsequent revisions of the paper, we plan to provide a more detailed derivation of it in appendix.
>
>
> > **Q4**: Lemma 1 actually does not say much useful for the approximation...
>
> **A4**: The role of Lemma 1 is to tell readers 1) what is the essential optimization goal of the expected gradient difference method we proposed.  Because, in CAT scenarios, we cannot get the labels of all samples like the traditional coreset problem. 2) It is an important theoretical basis of Theorem 1 (line 188 in main paper). Like the previous CAT optimization works [1][2], the approximation of the problem itself is unable to provide an approximation error. Fortunately, the ultimate goal of the paper is to obtain a theoretically guaranteed ability estimate (i.e., Theorem 1), which provide detailed analysis of approximation error in appendix.
>
>
> > **Q5**: What distribution does $y$ come from?
>
> **A5**: As stated on line 173, "the normed gradient difference is calculated as an expectation $\mathbb{E}_{y \sim p _{\theta^t }}$ over the possible labelings, since student's response labels $y$ to the candidate questions are unknown in the selection step.". The distribution $y$ determined by the current estimate $\theta^t$. For clarity, we will hightlight this part in the next version.
>
>
> > **Q6**: Why the ''probability distance'' $H_p(\theta^{t},\theta^*)$ in (10) is not zero when $\theta^{t}=\theta^*$?
>
> **A6**: $H_p(\theta^{t},\theta^*)=E_{(q,y)\sim p_{\theta^t}}  [1/p_{\theta^*}(q,y)]$ in the upper bound measures the distance between $p_{\theta^t}$ and $p_{\theta^*}$, but it is different from the traditional probability distance (such as KL): The probability $p_{\theta}\in[0,1] \to  1/p_{\theta} >=1$ which determines that the minimum value of $H_p$ cannot be 0 (its theoretical minimum value is 1. More details about $H_p$ can be found in Section C of Supplementary Material (it can be assumed that $p_{\theta^t}$ is a delta function)
>
> > **Q7**: Can you provide the variance of your experimental results?
>
> **A7**: Sure. In our experiments, we repeated 10 times under different seeds. For deep learning methods (BOBCAT and NCAT), we trained 10 models under different seeds. Due to the space limit in this Rebuttal editor pane, we provide the variance results in the above global Rebuttal pane (pdf) for every reviewers to check.
>
>
> Reference:
>
> [1] Ghosh, Aritra, and Andrew Lan. "BOBCAT: Bilevel Optimization-Based Computerized Adaptive Testing." International Joint Conference on Artificial Intelligence. 2021
>
> [2] Zhuang, Yan, et al. "Fully adaptive framework: Neural computerized adaptive testing for online education." Proceedings of the AAAI Conference on Artificial Intelligence. Vol. 36. No. 4. 2022.

---

> > ### Comment · Reviewer_DatR · 2023-08-12
> >
> > Thank you for your reply.
> >
> > **Regarding Q1 and my general opinion**\
> > I think my point was to ask adding $\forall \epsilon>0$, which is standard in a convergence proof. I think you got the point and this concern is relieved. Regarding the definition of $Q$, I meant there is no statistical definition of that, e.g., the distribution of $q\in Q$ and the true answers of $q$. Without making an assumption regarding the data generation process (DGP) of $q\in Q$, for sure the approximation error in Proposition 1 is inaccessible. I would suggest the author make reasonable assumptions that fit CAT about the DGP of $q$, and that gives the problem formation more structurality and hence deriving more rigorous results. I appreciate the authors' efforts in teaching me the background of CAT, and my ethical concern is dismissed. This is generally an interesting and insightful paper from my perspective and some insufficient mathematical rigor from the point of statistics is secondary to the main contribution, hence I am happy to increase my score to borderline accept.

---

> > > ### Author Response · Authors · 2023-08-12
> > > **Official Comment by Authors**
> > >
> > > Thank you for your response. We are appreciative of your thoughtful feedback and the time you've taken to reconsider the paper. Your suggestion to introduce reasonable DGP assumptions aligned with CAT is insightful. I see how this can contribute to a more structured problem formulation and lead to more robust outcomes.
> > >
> > > I'm glad that the paper's background on CAT is helpful and address ethical concerns. Your positive assessment of the paper as both interesting and insightful is truly motivating. We're committed to making the necessary improvements based on your feedback to enhance the overall quality of the paper.

---

### Official Review · Reviewer_GvrK · 2023-07-04

**Soundness:** 3 good
**Presentation:** 3 good
**Contribution:** 3 good
**Rating:** 6
**Confidence:** 4

**Summary:**

This paper proposes a method for computerized adaptive testing by selecting questions that have similar gradients in their expected responses to other questions in the question bank. Results show that this method works well on real-world datasets.

**Strengths:**

The proposed method seems sound. Experimental results suggest that it is effective

**Weaknesses:**

- The proposed method is largely unsurprising and I do not find it to be significantly different than existing active learning methods, except for a different application
- The performance improvement compared to existing methods is minimal

**Questions:**

- Can you clarify the significance of your theoretical analysis? They seem to be quite standard for submodular optimization. Strongly convex loss functions may not be feasible, as the authors mentioned, for more complex student models other than IRT. I also find the assumptions in Theorem 2, i.e., the minimized loss is 0, to be achievable. I the only case when that happens is the estimated ability parameter going to infinity (BCE minimization without regularization, so the MLE does not exist)?
- In my personal opinion, minimizing the gradient difference with other questions to select a most representative question is not that different than doing a training-test split, which does not make the proposed method significant different from existing work on using meta learning for CAT. Is the only reason about submodularity giving you a proof?
- Can you discuss what happens if the question bank is skewed in question parameters? In that case an approximation, Section 3.1, may not work?

---

> ### Author Rebuttal · Authors · 2023-08-08
>
> Thank you for your thoughtful comments and questions regarding our manuscript. Please feel free to share any further insights or suggestions you might have.
>
> > **Q1**: The proposed method is largely unsurprising and I do not find it to be significantly different than existing active learning methods.
>
> **A1**: In my opinion, regardless of CAT, Active Learning, Coreset, and Data Distillation, they are all sample selection or generation strategies (using the fewest samples to obtain the maximum benefit). Technically, their boundaries are not so clear. For example, the classic FSI in CAT selects a question whose difficulty is close to the ability (50% probability of correct answer), which is consistent with the uncertainty method in active learning. Recently, some researchers apply active learning to CAT [1]. This is an open-ended question, and we welcome any ideas you may have for discussion.
>
> > **Q2**: The performance improvement is minimal.
>
> **A2**: Compared to deep learning based methods, Yes, the improvement of our method BECAT at the beginning of the exam is small (line 265). Because the method based on deep learning requires pre-training selection algorithms on large-scale response data. However, information-based methods do not require training, thus it is difficult to catch up with the deep learning method. Considering the overhead and the bias from dataset, we still choose information-based paradigm. In the future, we will try to adapt the proposed explicit algorithm to data-driven frameworks.
>
> > **Q3**: Can you clarify the significance of your theoretical analysis?
>
> **A3**: Sure. The focus of this paper is: how to design a reasonable question selection algorithm, accurately estimating ability, when ground-truth ability is unknown. The significance of our theoretical analysis stems from our approach to problem identification, approximation and transformation, wherein we reframe the original problem into submodular optimization. Furthermore, from a technical standpoint, our work builds upon submodular and propose a novel similarity measurement method ($\widetilde{w}(i,j)$), which can be practically applied to CAT scenarios, and provide theoretical analysis (estimation error upper-bound).
>
> > **Q4**: Strongly convex loss functions may not be feasible, as the authors mentioned, ...other than IRT.
>
> **A4**: Yes, it is almost impossible to achieve strong convexity for a more complex user model. Luckily, IRT is the most widely used and most interpretable model in CAT. Recently, researchers use more complex cognitive diagnostic models (e.g. NeuralCDM) in CAT system. However, like related theory research, we cannot prove Theorem's effectiveness on a black-box neural network, so we only evaluate them in experiments as stated in line 200.
>
> > **Q5**: I also find the assumptions in Theorem 2, i.e., the minimized loss is 0, to be achievable when that happens is the estimated ability parameter going to infinity?
>
> **A5**: For all responses correct or all incorrect, no finite ML/BCE estimates exist [2] just as you said. Generally: given 2PL-IRT model: $p_j(\theta) = sigmoid(a_j(\theta-b_j))$, if a student correctly answers question A (with difficulty $b= 1.0$), but wrong answers question B (with difficulty $b= 3.0$), it seems that the ability estimate $\hat{\theta} = 2.0$ is a good choice. Moreover, the discrimination parameter $\alpha$ will also affect such estimation: for a question with high $\alpha$, the ability slightly greater than difficulty can make the probability approach 1 (loss=0). Then again, loss is 0 in Theorem 2 is relatively ideal. But its purpose is to illustrate that $H_p$ is small in upper bound (line 207). Therefore, we further provide experimental results in Appendix E.3, and it is found that $H_p$ can be maintained at a relatively small value.
>
> > **Q6**: In my personal opinion, minimizing the gradient difference...from existing work on using meta learning for CAT. Is the only reason about submodularity giving you a proof?
>
> **A6**: We understand your concern and we would like to emphasize that our method leverages the concept of minimizing gradient differences to identify a question that maximizes **information gain**. Moreover, compared with meta learning, this greedy question selection algorithm does not need to be pre-trained with large-scale response data. Regarding your question about submodularity, it is indeed an important factor in providing a rigorous proof. No theoretical guarantee is unacceptable for real-world standardized tests, which is why we propose an explicit selection algorithm (line99). But our contributions extend beyond this aspect, and the submodularity in the context of adaptive testing is novel and has the potential to advance the field by ensuring the optimality of question selection.
>
> > **Q7**: Can you discuss what happens if the question bank is skewed in question parameters? In that case an approximation, may not work?
>
> **A7**:  Item parameter estimation needs to be pre-calibrated before testing, which is crucial in CAT system, regardless of the selection algorithm. For example, when the difficulty parameter of a difficult question is wrongly estimated to be small, it will undoubtedly bring challenges to low-ability student. We acknowledge that if the question bank exhibits significant skewness, the approximation might face challenges in accurately estimating ability. To address this concern, we plan to extend our discussions and experiments to include a thorough analysis when dealing with skewed question parameter distributions.
>
> Reference
>
> [1] Bi, Haoyang, et al. "Quality meets diversity: A model-agnostic framework for computerized adaptive testing." 2020 IEEE International Conference on Data Mining (ICDM). IEEE, 2020.
>
> [2] Van der Linden, Wim J., and Peter J. Pashley. "Item selection and ability estimation in adaptive testing." Elements of adaptive testing. New York: Springer New York, 2009. 3-30.

---

> > ### Comment · Reviewer_GvrK · 2023-08-14
> >
> > thank you for a detailed response which clarified things a little bit. my opinion has not really changed; the paper seems technically solid, although I personally did not get many insights out of it.

---

> > > ### Author Response · Authors · 2023-08-15
> > > **Official Comment by Authors**
> > >
> > > We understand that the insights you were looking for might not have been fully met by our work. We value your perspective and will consider your feedback as we continue to improve our research. If you have any further questions or suggestions, we would be eager to hear them.
> > >
> > > Once again, thanks for your engagement with our paper.

---

### Official Review · Reviewer_TJRB · 2023-07-05

**Soundness:** 3 good
**Presentation:** 3 good
**Contribution:** 4 excellent
**Rating:** 8
**Confidence:** 5

**Summary:**

This paper tries to answer a question in computerized adaptive testing: how to select a question suitable for student without knowing the ground-truth of his/her true ability. To this end, the authors find the theoretical approximation of the true ability and provide theoretical and experimental analysis to support their proposition. They further develop an expected gradient difference approximation to design a greedy selection algorithm, successfully bounding the estimation error. Extensive experimental results show the estimation efficiency and accuracy of the method compared with baseline systems.

**Strengths:**

S1. An explicit question selection algorithm is proposed, which theoretically solves the dilemma that previous work can only approximate the truth ability implicitly. The solution is interesting: it finds a theoretical approximation of the true that is easily overlooked: the ability estimated by full responses to question bank. Moreover, this paper provides a convincing analysis of the plausibility of the approximation, both experimentally and theoretically.

S2. In terms of technical implementation, this paper improves the Coreset methods and propose an expected gradient difference approximation for CAT scenario. The authors further design a simple but effective greedy selection algorithm, and prove the error upper-bound of the ability estimation on questions found by it.

S3. Experiments on both real-world and synthetic datasets, show that it can reach the same estimation accuracy using 15% less questions on average, reducing test length.

S4. This paper is easy to follow. The additional material covers the proof and almost all additional experiments, including the detailed experimental analysis of Theorem in the paper.

**Weaknesses:**

1. The expected gradient difference approximation proposed in this paper is general in my opinion, and should not be limited to the CAT scenario. The sample selection strategy design or the efficient learning in the parameter estimation scenario seems to be applicable. I suggest the authors to study the effectiveness of the method in more tasks/domains.

2. As stated in the paper: "BECAT cannot surpass all other methods at the beginning of the exam". This is a cold start topic in an educational recommendation/testing. The reason is not explained in detail. Why the proposed method cannot surpass the RL/Meta learning methods to solve this cold start problem.

3. I have some questions about Theorem 2: What does the H_p function measure? What is the relationship between it and the target in this paper (true ability) and the error upper bound in Theorem 1?

**Questions:**

See above.

**Limitations:**

The limitations are clearly stated in this paper and its supplementary material.  It seems longer to train than traditional methods, but it adopts two speed-up tricks to improve the complexity.

---

> ### Author Rebuttal · Authors · 2023-08-09
>
> We greatly appreciate your thoughtful insights and suggestions regarding the generality of the expected gradient difference approximation proposed in our paper. Your perspective on the broader applicability of this approach is indeed intriguing and aligns well with our intentions.
>
> > **Q1**: The expected gradient difference approximation proposed in this paper is general in my opinion, and should not be limited to the CAT scenario.... I suggest the authors to study the effectiveness of the method in more tasks/domains.
>
> **A1**: We agree with your viewpoint that the concept of minimizing gradient differences has the potential to extend beyond the realm of Computerized Adaptive Testing. The proposed sample selection strategy design and efficient learning techniques for parameter estimation scenarios hold promise for a wide range of applications. In light of your recommendation, we are excited to investigate the feasibility and effectiveness of our approach in diverse contexts.
>
>
> > **Q2**: As stated in the paper: "BECAT cannot surpass ... Why the proposed method cannot surpass the RL/Meta learning methods to solve this cold start problem.
>
> **A2**: Thank you for your valuable feedback on our paper. The observed limitation in BECAT's early performance can be attributed to this key factor: CAT begins with no information about the student's abilities. Unless a fixed strategy is employed, the first question is usually chosen randomly, which can undoubtedly influence the information-based algorithms, like FSI, KLI and our BECAT. In contrast, RL/Meta learning methods have the potential to alleviate this challenge to a certain extent due to their ability to discern patterns by pre-training on large response dataset. Considering the training overhead in practice and the bias from dataset, we still choose information-based paradigm in this paper. By addressing this aspect, we aim to provide a more comprehensive analysis of BECAT's performance in different phases of an exam.
>
> > **Q3**: I have some questions about Theorem 2: What does the $H_p$ function measure? What is the relationship between it and the target in this paper (true ability) and the error upper bound in Theorem 1?
>
> **A3**: As stated in line 202 of the paper, $H_p(\theta^{t},\theta^*)$ can be regarded as a type of statistical distance: measuring how probability distribution $p_{\theta^t}$ is different from $p_{\theta^*}$. Moreover, with the help of the consistency estimation (i.e., binary cross-entropy) at each step, $H_p(\theta^t,\theta^*)$ can reach its theoretical minimum. Theorem 1 suggests that, to minimize the expected error bound, the CAT systems should try to minimize $H_p$. In this case, the estimate error upper bound in Theorem 1 can be as small as possible. In addition, we further provide experimental results in Appendix E.3, and it is found that $H_p$ can be maintained at a relatively small value.

---

> > ### Comment · Reviewer_TJRB · 2023-08-15
> > **Thanks for the authors' responses**
> >
> > Thank you for the authors' detailed responses.
> >
> > Most of my concerns have been thoroughly addressed. In particular, the explanations regarding Theorems 1 and 2, especially their connection to the overall objectives, are now clear to me.
> >
> > I would appreciate additional insight on this. Additionally, I'm interested in the practical implementation of your proposed method. Are there specific considerations concerning computational resources or model complexity that users should note?
> > I would like to hear about it in some details.

---

> > > ### Author Response · Authors · 2023-08-16
> > > **Official Comment by Authors**
> > >
> > >
> > >
> > > Thank you for your positive feedback. We appreciate your recognition of our efforts to address your concerns and provide clarity on the key aspects of our work.
> > >
> > > Thank you for your follow-up questions. As described in Appendix D, these two tricks successfully reduce our method from $O(|Q||\Theta|)$ to $O(|Q|m)$, $m \ll |\Theta|$, and greatly reduce the calculation frequency in selection (Figure 3 in Appendix shows the improved algorithm efficiency). Besides, the proposed selection method requires no additional training, not even GPU resources, compared to recent deep learning methods.
> > >
> > > If you have any other questions, please feel free to ask.

---

> > > > ### Comment · Reviewer_TJRB · 2023-08-17
> > > >
> > > > Thank you for addressing practical implementation considerations. Ensuring informative and efficient question selection is good.
> > > > All my concerns have been addressed. I am happy to increase my score. I would encourage the author to include the information we discussed in the further paper.

---

> > > > > ### Author Response · Authors · 2023-08-18
> > > > > **Official Comment by Authors**
> > > > >
> > > > > Thank you for your time and thoughtful evaluation. It's great to hear that all your concerns have been successfully addressed.
> > > > >
> > > > > Your insights and suggestions are valuable to us, and we will include the discussed information in the future version.

---

### Official Review · Reviewer_KCbW · 2023-07-06

**Soundness:** 3 good
**Presentation:** 1 poor
**Contribution:** 3 good
**Rating:** 6
**Confidence:** 3

**Summary:**

This paper investigates the problem of effective question selection in Computerized Adaptive Testing (CAT), with the goal of designing a procedure that minimizes test length while maximizing  estimation accuracy of student ability. While student ability doesn’t have a known ground-truth, they use the student ability estimated using all questions (as opposed to a small subset) as the effective ground truth. To design a short test, they use literature on core set selection to pick a subset of questions that have the same gradient updates as if using the entire question bank.  This scheme is used in some real world experiments, demonstrating gains in shorter test lengths and better estimation accuracy.

**Strengths:**

I generally like the results in this paper and I believe the CAT community would find this to be a valuable contribution.

- As far as I am aware, the ideas in this paper are novel. It proposes a simple, computationally inexpensive approach to question selection. The work nicely combines existing literature in diverse areas of AI to solve a well-defined problem.

- The results in this paper are compelling and compared against a thorough set of competitive models from recent literature. The paper’s approach is much more computationally simple than that of these competitors, and seems to empirically outperform them.

**Weaknesses:**

My biggest critique of this paper has to do with its writing and presentation, which I would say needs a lot of work. While I find the results interesting, the presented methodology is hard to follow and filled with errors and typos. Just a few examples:

- What does the ⇒ in equation 5 mean? This is not a gramatically well-formed mathematical sentence.
- In equation 5, what is $d$? I don’t think this is defined anywhere
- In the simulation experiment, the paper simulates student abilities “using the smallest EXAM dataset”. What does that mean exactly? Do you also simulate the question parameters or estimate those first? I think this is explained in appendix E, but very confusing when you just read the main paper. Relatedly, in task 2, do you mean $\theta^*$ instead of $\theta_0$ (which is unknown)

Related to the writing, I found the math exposition in this paper to be extremely difficult to follow. Many of the steps are explained poorly, making them hard to verify (e.g. Lemma 1 in Appendix A).

**Questions:**

For the ability estimated by the responses to entire question bank to actually converge to the true ability (as shown by your synthetic experiments in Fig 2.) I believe you need the ability parameter to be static over time. If i remember correctly, the EEDI data is collected over a long period of time, so students likely (hopefully) are learning over time. Is there any way to address this issue?

**Limitations:**

Right now it sounds like one of the paper’s contributions is the idea of using the entire set of questions to represent the unknown student ability. However, this is something also explored in the BOBCAT paper. I would recommend citing them and carving out your contribution on top of this. For ex. “we are the first to propose a selection algorithm that explicitly targets this full sample estimate”.

---

> ### Author Rebuttal · Authors · 2023-08-07
>
> We would like to express our sincere gratitude for your high appreciation of the contribution and novelty presented in our paper. Your positive feedback means a lot to us. We also appreciate your valuable suggestions regarding the refinement of certain technical aspects in the paper to enhance clarity. Following your advice, we will further emphasize the difference from other literature (e.g., BOBCAT), and on top of it, explicitly highlight our contribution in the paper.
> Here are the responses to each of your questions:
>
>
> > **Q1**: What does the "$\Rightarrow$" in equation 5 mean? This is not a gramatically well-formed mathematical sentence.
>
> **A2**: The "$\Rightarrow$" is used because we cannot directly solve the original problem in Equation 5 and need some approximation to it. For rigor, we use "$\Rightarrow$" to indicate such transformations. Sorry to cause confusion to you, we will change it in the next version.
>
>
> > **Q2**: In equation 5, what is $d$? I don’t think this is defined anywhere.
>
> **A2**: Sorry for missing this definition in the main paper: $d=\max_{i\in Q,j\in S}\max_{\theta\in \Theta} {\Vert \nabla l_i(\theta) - \nabla l_{j}(\theta)\Vert}$ is the maximum pairwise gradient distance.
>
> > **Q3**: In the simulation experiment, ..."using the smallest EXAM dataset". What does that mean exactly? Do you also simulate the question parameters or estimate those first?
>
> **A3**: Since the bank size $|Q|$ can impact Proposition 1 (the larger the $Q$, the better the approximation), we selected the smallest dataset EXAM (with smallest $Q$) to verify this proposition. We use the training set of real data to estimate the item parameters and fix them in the experiments. Due to space limits, we simplify this part in the main paper.
>
>
> > **Q4**: In task 2, do you mean $\theta^*$ instead of $\theta_0$ (which is unknown).
>
> **A4**: Task 2 is a simulation experiment, which needs to manually generate some $\theta_0$s. In order to make these $\theta_0$s conform to the students' ability distribution, we use all the students' responses in the dataset to estimate their ability $\{\theta_0^1,\theta_0^2, ...,\theta_0^N\}$ as the ground truth $\theta_0$ like [1][2].
>
> > **Q5**: Many of the steps are explained poorly, making them hard to verify (e.g. Lemma 1 in Appendix A).
>
> **A5**: Some steps are skipped is mainly because some of these proof steps (e.g., Lemma 1) are based on the other papers, which have been cited in the proofs. However, the proofs of the Theorem originally proposed in our paper such as Theorem 1 and 2 are detailed.
> Thanks for your suggestion, we promise to provide a detailed proof of Lemma 1 in an improved version of the manuscript. (*Detailed proofs of Lemma 1 can now be found in the Global Rebuttal pane, for all reviewers to check.*)
>
> > **Q6**: For the ability estimated by the responses... I believe you need the ability parameter to be static over time. ...so students likely are learning over time. Is there any way to address this issue?
>
> **A6**: In fact, as you said, student abilities that are not static in most public datasets. However, these datasets look like "session-based", i.e., the time of response behavior in a session is compact (minutes or hours), and the interval between sessions is large (days or even months). In a session, students' ability will not change much, so a feasible method is: divide a student response sequence into different sections according to the time feature/session, and treat each section as a completely different student. In addition, we collected a student exam dataset EXAM in our paper, which collected the records of junior high school students on mathematical exams. We will make this dataset publicly available after the paper is accepted.
>
>
> Reference:
>
> [1] Bi, Haoyang, et al. "Quality meets diversity: A model-agnostic framework for computerized adaptive testing." 2020 IEEE International Conference on Data Mining (ICDM). IEEE, 2020.
>
> [2] Cheng, Ying. "When cognitive diagnosis meets computerized adaptive testing: CD-CAT." Psychometrika 74 (2009): 619-632.

---

> > ### Comment · Reviewer_KCbW · 2023-08-12
> >
> > Thanks for your answers, they were quite helpful.
> >
> > As mentioned in my initial review, I like this paper and think it would be a valuable contribution to the CAT community. I strongly urge the authors to spend a lot more time or get external advice on the writing and presentation.
> >
> > I will maintain my initial rating of a weak accept because I'm not sure if NeurIPS is necessarily the right place for this paper to reach the relevant audiences and I still have concerns about writing and clarity.

---

> > > ### Author Response · Authors · 2023-08-14
> > > **Official Comment by Authors**
> > >
> > > Thank you for your time and valuable insights. Your acknowledgment means a lot to us, and we are honored to contribute meaningfully to the CAT community through our paper. Thank you for your suggestions about presentation, and we will further optimize the paper, making it accessible to general NeurIPS readers in other fields. This is also important for broadening the impact of CAT field itself in NeurIPS and beyond.

---

### Official Review · Reviewer_aSHy · 2023-07-10

**Soundness:** 2 fair
**Presentation:** 2 fair
**Contribution:** 3 good
**Rating:** 4
**Confidence:** 3

**Summary:**

The authors advocate to propose a method to better estimate students ability by using as few questions as possible. They redefine Computer Adaptive Testing(CAT) as a adaptive subset selection of question to estimate students ability and propose a gradient based selection method to select items that minimizes the estimation error term. Through experiment results they show that their proposed method BECAT outperforms existing  CAT method at reducing test length by 10-20%.

**Strengths:**

The author study an important problem of how to efficiently estimate the ability of student by adaptively asking them as small number of questions as possible. They propose a gradient based method to solve this problem and their empirical results suggest that their method perform superior to other CAT methods at reducing test length. Empirical results suggest that their method perform superior to other CAT methods at reducing test length.

**Weaknesses:**

1. The paper is poorly written and is not easy to follow.
2. Important concepts and definitions like student’s “ability”, model of the student is not well defined. For more details refer to the question section.
3. Even through the paper studies an important problem, the overall presentation and readability of the paper can be significantly improved. At this point I cannot propose to accept this paper.

**Questions:**

1. “For accurate and efficient assessment” : how is ground truth “ability” $\theta_0$ defined mathematically? what is $\Theta$ set mathematically?

2. How can the ability be estimated by minimizing the empirical loss? Can the authors provide a concrete example?

3. In line 82-83, what is “standard” gradient descent and under what ability space $\Theta$ the gradient descent can minimize empirical risk

4. The student’s ability is defined to be fixed $\theta_0$ in line 75, and in line 90 student current ability is said to be $\theta^t$ which is an evolving parameter through time. How are these two definitions consistent?

5. “there is no such ground truth $\theta_0$ in the dataset” what does a parameter being in a dataset even mean?

**Limitations:**

The cognitive model of student is over simplied.

---

> ### Author Rebuttal · Authors · 2023-08-04
>
> Thank you for your valuable feedback. We understand that there might be room for improvement in terms of clarity. Regarding the questions you raised, we have carefully considered each point and have made following responses:
>
> > **Q1**: How is ground truth "ability" $\theta_0$  and $\Theta$ defined mathematically? Student’s “ability”, model of the student is not well defined.
>
> **A1**: In Item Response Theory (IRT), the student's ability parameter $\theta$ refers to the latent trait being measured. It represents the underlying ability of the student that is not directly observable but influences their performance on the test items [1]. Therefore, $\theta_0$ is a student's true ability parameter, which can be estimated by student's performance on the test items. As stated in line 36, you can simply think of it as a parameter estimation problem: $\theta_0$ is the unknown truth value to be estimated, $\Theta$ is the entire parameter space, and the student's performance is the observation $X$.
>
> > **Q2**: How can the ability be estimated by minimizing the empirical loss? Can the authors provide a concrete example?
>
> **A2**: In Item Response Theory (IRT), the ability of a student can be estimated by minimizing the empirical loss when using a probabilistic model such as the logistic model for binary responses. Minimizing the empirical loss involves finding the ability parameter $\theta$ that best fits the observed response data [2]. To illustrate with a concrete example (line 300), let's consider a simplified version of the 1-parameter logistic (1PL) IRT model for binary responses. In this model, the probability of a correct response to item $i$ for a student with ability $\theta$ is given by:
>
> $
> P(\text{Correct response to item } i) = \frac{1}{1 + e^{-(\theta - b_i)}}.
> $
>
> where $b_i$ is the difficulty parameter for item $i$ (the ability level at which the item has a 50% chance of being answered correctly). Now, assume we have a dataset with binary responses (0 for incorrect, 1 for correct) for a set of items administered to multiple students. The empirical loss function, such as Binary Cross Entropy (BCE) loss, can be used to estimate the ability parameter $\theta$. We can use optimization techniques like gradient descent or other numerical methods to find the minimum of the empirical loss function. The estimated $\theta$ will be the ability parameter that best fits the observed data based on the IRT model.
>
> >  **Q3**: In line 82-83, what is “standard” gradient descent?
>
> **A3**: Standard Gradient Descent, also known as Batch or Deterministic Gradient Descent, is an optimization algorithm that use the **entire** training set. It process all the training examples simultaneously in a large batch [3]. This terminology can be somewhat confusing because the word “batch” is also often used to describe the minibatch used by minibatch stochastic gradient descent. Typically the term “standard/batch gradient descent” implies the use of the full training set, while the use of the term “batch” to describe a group of examples does not. Therefore, to prevent unnecessary misunderstandings caused by "batch" to readers, we use "standard" instead in this paper.
>
> > **Q4**: Under what ability space $\Theta$ the gradient descent can minimize empirical risk?
>
>
>
> **A4**: From the perspective of optimization, it is not fundamentally different from general gradient descent problems, as it requires finding an optimal value $\theta^*$ in the whole parameter space $\Theta$.
>
> > **Q5**: The student’s ability is defined to be fixed $\theta_0$ in line 75, and in line 90 student current ability is said to be $\theta^t$ which is an evolving parameter through time. How are these two definitions consistent?
>
> **A5**: Since students do not receive correctness feedback during the test, the true value of the student's ability, $\theta_0$, is **constant** and **unchanged** throughout the test. As mentioned in Section 1 of the paper, current ability $\theta^t$ is an "estimate", and CAT needs to sequentially use student responses to keep this estimate $\theta^t$ close to $\theta_0$. Therefore, CAT is essentially an "online" parameter estimation problem, and our contribution is to design a reasonable question selection strategy for efficient estimation, when the groundtruth $\theta_0$ is unknown.
>
> > **Q6**: "there is no such ground truth $\theta_0$ in the dataset" what does a parameter being in a dataset even mean?
>
> **A6**: As stated in line 33 of the paper, CAT hopes that the question selection algorithm we designed can select the fewest suitable items to make the ability estimation more accurate. In essence, it is trying to solve the optimization problem in Definition 1: $\mathop{\mathrm{min}}\limits_{|S|=T}	\Vert {\theta^T}-\theta_0 \Vert$. However, this optimization problem **cannot** be solved explicitly due to the unknown of ground-truth $\theta_0$ in the dataset. This is why we use $\theta^*$ to approximate $\theta_0$ and reformulate such problem (i.e., Definition 2).
>
>
> > **Q7**: The cognitive model of student is over simplied.
>
> **A7**: In fact, the student model in traditional CAT and psychometrics is such a "simple". The most widely used user model in CAT is IRT, which is usually a form of logistic regression. Meanwhile, the cross-entropy loss of the L2-regularized IRT is strongly convex, which is the assumption of Theorem 1. In recent years, more "complex" models have emerged, such as NeuralCDM, which uses neural networks to model students. Like optimization theory research, we cannot prove the effectiveness of Theorem on a black-box neural network, so we evaluate it in experiment part.
>
>
>
> Reference:
>
> [1] Embretson, Susan E., and Steven P. Reise. Item response theory. Psychology Press, 2013.
>
> [2] Baker, Frank B., and Seock-Ho Kim, eds. Item response theory: Parameter estimation techniques. CRC press, 2004.
>
> [3] Goodfellow, Ian, Yoshua Bengio, and Aaron Courville. Deep learning. MIT press, 2016.

---

> > ### Comment · Reviewer_aSHy · 2023-08-18
> >
> > Thanks for your response! I still think the clarity of the paper can be further improved by using proper notations and making the testing setup more clear. For example, it would help to clarify the mathematical model from which the sample $(q_t, y_t)$ have been assumed to be generated from. Regarding notation, in line 110, I am not sure why we need limit why not say $\theta^* = \theta^{|Q|}$ when $|Q|$ is a fixed positive integer. Further, if question bank size is fixed, the learner would not gain anything by seeing repeated questions. In that case, the limit in $\theta^* \approx \lim_{t\rightarrow \infty} \theta^t \approx \theta_0$ does not make much sense. Further, I would appreciate a discussion section on 'true' ability of the student mathematically and if there would be any approximation error when one tries to approximate the 'true' ability by parametrized family $\Theta$. For now, I would keep my score.

---

> > > ### Author Response · Authors · 2023-08-18
> > > **Official Comment by Authors**
> > >
> > > Thank you for your follow-up questions! We are grateful for your commitment to helping us improve the paper. The following are responses to your questions.
> > >
> > >
> > > > **Q1**: The mathematical model from which the sample $(q_t, y_t)$ have been assumed to be generated from.
> > >
> > > **A1**: Since the ground truth of student ability $\theta_0$ is not available, a feasible and commonly used evaluation method in CAT field is simulation experiment (Task 2) [1]. Specifically, we artificially generate their $\theta_0$ and further simulate student-question interaction process within CAT systems (line 223).
> > >
> > > Let me give a concrete example: suppose the student model is 2PL-IRT: $p_j(\theta)=sigmoid(\alpha_j(\theta - b_j))$, where each question $j$ is characterized by two parameters: the discrimination parameter ($\alpha_j$) and the difficulty parameter ($b_j$). To generate the true ability values ($\theta_0$) for our simulated $N$ students, we can sample $\\{\theta_0^1, \theta_0^2,...\theta_0^N\\}$from a known distribution, such as a normal distribution. These ability values represent the latent trait being measured by the test questions. Therefore, given one generated student $i$ whose ability is $\theta_0^i$, for any question $q_k$ selected by CAT, the probability $p_k(\theta_0^i)$ of the correct response can be calculated by the above IRT, and the corresponding response label $y$ can be generated according to the threshold (e.g., 0.5).
> > >
> > > > **Q2**: Why we need limit why not say $\theta^*=\theta^{|Q|}$ when $|Q|$ is a fixed positive integer.
> > >
> > > **A2**: Yes, $\theta^*=\theta^{|Q|}$ as you said, because $\theta^*$ is the estimate which is "estimated by his/her full responses to the entire bank $Q$" (Proposition 1). The reason why we use the equivalent limit form ($\theta^*=\lim\limits_{t\to|Q|}{\theta}^t$) in the proof is to prove: $\theta^* \approx \lim\limits_{t\to \infty}{\theta}^t \approx \theta_0$ ($t\in[0,|Q|]$), which also has a limit form. We would have wanted the reader to understand our proof more intuitively. Thanks for your feedback, we will improve this notation in the future to avoid confusion.
> > >
> > >  > **Q3**: if question bank size is fixed, the learner would not gain anything by seeing repeated questions. In that case, the limit in $\theta^* \approx \lim\limits_{t\to \infty}{\theta}^t \approx \theta_0$ does not make much sense.
> > >
> > > **A3**: In general machine learning problem, the consistency property [2] in maximum likelihood estimation (i.e., $\lim\limits_{t\to\infty}p\left(\left|{\theta}^t-\theta_0\right|\geq\epsilon\right)=0$) means: *when the number of observed samples $t$ is huge (infinity), the estimated value $\theta^t$ is almost the same as the true value $\theta_0$*. An implicit premise of it is that the samples must be (mostly) **diverse**. Otherwise, if the 60,000 handwriting images in the MNIST dataset are all the same, the training result must be bad undoubtedly as you said. Therefore, in the corresponding CAT scenario, the meaning of $t\to \infty$ is essentially that student needs to response to as many (and diverse) questions as possible. This requires the question bank $Q$ to approach infinity. To verify this, we use simulation experiments (line114): Figure 2(a) shows that when the bank size exceeds 300, the estimated $\theta^* \approx \theta_0$. (By the way, the questions that have been selected need to be removed from the candidates)
> > >
> > >
> > > We truly appreciate your suggestion, and we will include the details of our discussion above (e.g., mathematical definition of student true ability) in a future version. Please feel free to share any further insights or suggestions you might have.
> > >
> > >
> > > Reference:
> > >
> > > [1] Vie J J, Popineau F, Bruillard É, et al. A review of recent advances in adaptive assessment[J]. Learning analytics: Fundaments, applications, and trends: A view of the current state of the art to enhance e-learning, 2017: 113-142.
> > >
> > > [2] Eliason S R. Maximum likelihood estimation: Logic and practice[M]. Sage, 1993.

---

> ### Comment · Reviewer_KCbW · 2023-08-12
>
> I will corroborate the authors that the cognitive model used in this paper is standard in the testing literature and is used universally in practice e.g. in the GRE, Deep Knowledge Tracing, learning apps, etc.

---

### Author Rebuttal · Authors · 2023-08-07

We appreciate all the reviewers' thorough assessment and valuable feedbacks. Their thoughtful evaluation has provided valuable insights that have significantly contributed to the improvement of the manuscript. The reviewers' positive comments encompassing different dimensions are truly encouraging:

- **Contribution**:
"CAT community would find this to be a valuable contribution" (KCbW);
"the ideas in this paper are novel" (KCbW);
"This is a very interesting paper/solution is interesting" (TJRB, DatR);
"The author study an important problem" (aSHy);

- **Presentation**:
"This paper is easy to follow" (TJRB);
"The proposed algorithm logically makes sense to me, and the English writing is generally clear enough" (DatR);
"The organization of the paper is good, which helps me understand the main content quickly" (DatR);


- **Method**:
"It proposes a simple, computationally inexpensive approach" (KCbW);
"this paper provides a convincing analysis of the plausibility of the approximation" (TJRB);
"coreset finding in efficient ability estimation for CAT is promising" (DatR)


- **Experimental Results**:
"their method perform superior to other CAT methods at reducing test length" (aSHy);
"The results are compelling and compared against a thorough set of competitive models" (KCbW);
"this method works well on real-world datasets" (GvrK)

Regarding the questions raised by each reviewer, we have carefully considered each point and have made detailed responses in the local rebuttal pane.

---
**[Response to Ethics Reviewers]**

We deeply appreciate your thoughtful consideration of the ethical implications associated with CAT itself. Like recommendation system (different users will be recommended different items), CAT is also a **personalized** question in Education and has fairness issues of course, which is another interesting topic. You point out that the use of adaptive methods within the standardized testing is a topic of ongoing debate, with concerns about bias in original data and its potential amplification during the estimation process. We acknowledge the importance of addressing these concerns to ensure fairness and equity in educational assessments.

While our paper aimed to present a novel method for improving the **efficiency** of CAT, your feedback highlights the necessity of addressing the potential ethical pitfalls. In line with your recommendation, we will revise the paper to incorporate a dedicated subsection that discusses potential sources of bias in CAT itself and the datasets used.

***

Several steps of Lemma 1 are skipped in the paper, because these steps are based on the other papers (which have been cited in the original proofs). Thanks for the suggestion from Reviewer KCbW, we now provide detailed proof of Lemma 1 for all reviewers to check:

**[Detailed proof of Lemma 1]**
> *Lemma1*: ...The corresponding designed selection algorithm using submodular function $\widetilde{F}$ is actually approximately solving the following optimization problem: $\min\limits_{|S|=T} \max\limits_{\theta\in \Theta} \mathbb{E}_y [\Vert  \sum _{j\in S}{\gamma_j \nabla l_j(\theta)} - \sum _{i\in Q}{\nabla l_i(\theta)}  \Vert ]$

 *Proof*:  We first define a mapping function $h$ from set $Q$ to $S$ to a mapping function:$\forall i \in Q, h(i)\in S$. It assign every response data point $i \in Q$ to one of the elements $j$ in $S$. Then, for any arbitrary ability parameter $\theta \in \Theta$ we can write
  $$ \sum_{i\in Q}{\nabla l_i(\theta)} =\sum_{i\in Q}{[\nabla l_i(\theta)- \nabla l_{h(i)}(\theta) +  \nabla l_{h(i)}(\theta)]}
	=\sum_{i\in Q}{[\nabla l_i(\theta) - \nabla l_{h(i)}(\theta)]} + \sum_{j\in S}{\gamma_j \nabla l_j(\theta)}$$
Subtracting and taking the expected norm of the both sides, we get an upper bound on the error. According to the triangle inequality, we have $$\mathbb{E}\left[\Vert \sum_{i\in Q}{\nabla l_i(\theta)} -\sum_{i\in S}{\gamma_j \nabla l_j(\theta)} \Vert \right]\le \sum_{i\in Q}{\mathbb{E}\left[\Vert \nabla l_i(\theta) - \nabla l_{h'(i)}(\theta)\Vert \right]}.$$
When the mapping function $h$ is to map each element in $Q$ to the one in $S$ that is closest to its expected gradient, the right side of inequality is minimized, or minimum expected distance between the gradient: $h(i)= \arg\min_{j\in S} \mathbb{E}\left[\Vert \nabla l_i(\theta) - \nabla l_j(\theta)\Vert \right]$. Therefore, the upper bound of the expected gradient difference can be further constrained:
$$\min_{|S|=T}\mathbb{E}\left[\Vert \sum_{i\in Q}{\nabla l_i(\theta)} -\sum_{i\in S}{\gamma_j \nabla l_j(\theta)} \Vert \right]\le \sum_{i\in Q}{\mathop{\mathrm{min}}\limits_{j\in S}\mathbb{E}\left[\Vert \nabla l_i(\theta) - \nabla l_{j}(\theta)\Vert \right]}.$$
Next, define a similarity function $\widetilde{w}(i,j)$ which measures the expected gradient similarity between response pair $i$ and $j$: $ \widetilde{w}(i,j)=d-\max_{\theta\in \Theta} {\mathbb{E}\left[\Vert \nabla l_i(\theta) - \nabla l_{j}(\theta)\Vert \right]} $, and $d=\max_{i\in Q,j\in S}\max_{\theta\in \Theta} {\Vert \nabla l_i(\theta) - \nabla l_{j}(\theta)\Vert}$ is the maximum pairwise gradient distance. Thus, the optimization problem can also be transformed as:
$$\max\limits_{|S|=T} \sum_{i\in Q}{\mathop{\mathrm{max}}\limits_{j\in S} \widetilde{w}(i,j)}.$$
Following the same way of origin problem, its corresponding submodular $\widetilde{F}(S)=\sum_{i\in Q}{\max_{j\in S} \widetilde{w}(i,j)}$, which is the same with our proposed method. Thus, the designed selection algorithm is the greedy algorithm of the optimization problem.


***

**[Variance results]**

Pdf is the variance result asked by Reviewer DatR, for every reviewers to check.

---

### Decision · Program_Chairs · 2023-09-21

**Decision:**

Accept (poster)

**Comment:**

The paper studies the problem of selecting informative questions in Computerized Adaptive Testing. The goal is to adaptively select a small number of questions for a student that will maximize the accuracy of estimating the student's ability. The proposed method, BECAT, is based on ideas from adaptive submodular optimization and operates by selecting a subset of questions that provides gradients similar to the entire set of questions. Extensive experimental evaluation on synthetic and real-world datasets showcases the effectiveness of the proposed method. The reviewers acknowledged that the paper considers an important problem setting in Computerized Adaptive Testing and that the proposed method has novel ideas with promising results. However, the reviewers and two Ethics reviewers raised several concerns and questions in their initial reviews. We want to thank the authors for their detailed responses and for actively engaging with the reviewers during the discussion phase. The reviewers appreciated the responses and have an overall positive assessment of the paper. The reviewers have provided detailed feedback in their reviews, and we strongly encourage the authors to incorporate this feedback as per the authors' responses. In particular, the authors should carefully incorporate the feedback from Ethics reviewers by adding a dedicated discussion subsection in the final version of the paper.